# ATP-independent substrate recruitment to proteasomal degradation in mycobacteria

Tatjana von Rosen[1], Monika Pepelnjak[2], Jan-Philipp Quast[2] , Paola Picotti[2], Eilika Weber-Ban[1]

**Mycobacteria and other actinobacteria possess proteasomal degradation pathways in addition to the common bacterial compartmentalizing protease systems. Proteasomal degradation plays a crucial role in the survival of these bacteria in adverse environments. The mycobacterial proteasome interacts with several ring-shaped activators, including the bacterial proteasome activator (Bpa), which enables energy-independent degradation of heat shock repressor HspR. However, the mechanism of substrate selection and processing by the Bpa-proteasome complex remains unclear. In this study, we present evidence that disorder in substrates is required but not sufficient for recruitment to Bpa-mediated proteasomal degradation. We demonstrate that Bpa binds to the folded N-terminal helix-turn-helix domain of HspR, whereas the unstructured C-terminal tail of the substrate acts as a sequence-specific threading handle to promote efficient proteasomal degradation. In addition, we establish that the heat shock chaperone DnaK, which interacts with and co-regulates HspR, stabilizes HspR against Bpa-mediated proteasomal degradation. By phenotypical characterization of *Mycobacterium smegmatis* parent and *bpa* deletion mutant strains, we show that Bpa-dependent proteasomal degradation supports the survival of the bacterium under stress conditions by degrading HspR that regulates vital chaperones.**

## Introduction

Proteome turnover, achieved through controlled protein degradation by compartmentalizing proteases, plays a crucial role in bacteria, serving as a mechanism for quality control and regulation. Bacteria use various chaperone–protease systems to help maintain cellular integrity (Laederach et al, 2014). In addition to these well-established bacterial degradation complexes, certain actinobacteria, including mycobacteria, have acquired a eukaryotic-like proteasome through horizontal gene transfer. Although non-essential under standard culture conditions, the proteasome supports survival of these bacteria under stress, for example, during the persistence of

the human pathogen *Mycobacterium tuberculosis* inside host macrophages (Darwin et al, 2003; Gandotra et al, 2007).

Similar to the eukaryotic 20S proteasome, the actinobacterial 20S proteasome forms a cylindrical particle composed of four stacked seven-membered rings. The two inner $\beta$-rings carry the protease active sites and the two gated outer $\alpha$-rings associate with ring-shaped activator complexes to recruit substrate proteins to the degradation chamber (Hu et al, 2006). For the degradation of folded cellular proteins, the 20S core particle (CP) interacts with a hexameric ATPase ring complex referred to as Mpa (mycobacterial proteasome ATPase) in mycobacteria or ARC (ATPase forming ring-shaped complexes) in other actinobacteria (Wolf et al, 1998; Darwin et al, 2005; Striebel et al, 2010).

Mpa associates with the 20S CP through a C-terminal GQYL motif, which shares a penultimate tyrosine with the C-terminal HbYX motif of the eukaryotic 19S ATPase subunits (Striebel et al, 2010; Kavalchuk et al, 2022). The coaxial alignment of the Mpa ring pore with the $\alpha$-ring pore of the 20S CP is further facilitated by dynamic, charge-driven interactions between the $\beta$-hairpin loops of Mpa's C-terminal $\beta$-grasp domains and helix H0 on the surface of the $\alpha$-rings (Kavalchuk et al, 2022).

The Mpa/ARC complex recognizes proteins that have undergone post-translational modification with the prokaryotic ubiquitin–like protein (Pup) in a process referred to as pupylation (Pearce et al, 2008; Striebel et al, 2010). Pup acts as a recruiter, binding to the N-terminal coiled-coil domains of the Mpa ring complex. Subsequently, Pup is engaged into the central pore of Mpa, leading to the unfolding and translocation of the pupylated substrate into the 20S proteolytic chamber in an ATP-dependent manner (Sutter et al, 2009; Wang et al, 2010). Pupylation-dependent proteasomal degradation has been well studied, and numerous pupylation substrates have been identified in mycobacteria (Festa et al, 2010; Poulsen et al, 2010; Watrous et al, 2010; Fascellaro et al, 2016).

The bacterial proteasome activator (Bpa) serves as an alternative ring-shaped homooligomer that interacts with the 20S CP (Delley et al, 2014; Jastrab et al, 2015; Bai et al, 2016; Bolten et al, 2016). Bpa uses the same C-terminal GQYL interaction motif as Mpa to associate with the 20S CP. Both Mpa and Bpa bind to the core particle by inserting the GQYL interaction motif into a binding pocket located between the $\alpha$-subunits of the 20S CP, effectively opening the

[1]ETH Zurich, Institute of Molecular Biology and Biophysics, Zurich, Switzerland   [2]ETH Zurich, Institute of Molecular Systems Biology, Zurich Switzerland

Correspondence: eilika@mol.biol.ethz.ch

gated channel to the proteolytic chamber (Bai et al, 2016; Bolten et al, 2016; Hu et al, 2018). However, apart from the C-terminal GQYL motif, Bpa does not exhibit any significant homology to Mpa. Bpa is a small, single-domain protein with a four-helix bundle fold. It assembles into a twelve-membered ring complex resembling a bowl that narrows from 40 Å inner diameter at the opening distal to the α-ring to an ~20 Å wide entrance pore on the side proximal to the α-ring interface.

Bpa lacks ATPase activity and was shown to degrade the non-globular protein β-casein, which serves as a model substrate for unfolded proteins because of its extended structure (Delley et al, 2014; Bolten et al, 2016). ATP-independent 20S proteasome activators are also found in eukaryotes (PA28 and PA200) and can stimulate ubiquitin-independent proteasomal degradation of small peptides or poorly folded substrates, including the unstructured τ protein (Huang et al, 2016). Consequently, an early hypothesis proposed that Bpa facilitates the proteasomal degradation of non-native or damaged proteins that may accumulate during environmental stress (Delley et al, 2014; Jastrab et al, 2015, 2017; Bolten et al, 2016).

To date, only two natural substrates have been identified for Bpa-mediated proteasomal degradation (Hu et al, 2018; Jastrab et al, 2015). One is the heat shock repressor HspR found in *M. tuberculosis* (Jastrab et al, 2015). HspR functions as a transcriptional repressor of the operon encoding the Hsp70/40 chaperone system (*dnaK, grpE, dnaJ*) and *hspR* itself. Its activity is supported by DnaK, which acts as a co-repressor by binding to HspR. During heat shock (42°C), HspR partially unfolds (Das Gupta et al, 2008; Bandyopadhyay et al, 2012), leading to its dissociation from DnaK and the specific DNA operator called HAIR (HspR-associated inverted repeats) motif (Bucca et al, 2000). This dissociation allows for the expression of vital heat shock chaperones, ensuring the maintenance of proteome integrity under heat stress conditions. However, the molecular mechanism underlying the recruitment of HspR and other potential substrates to Bpa-mediated proteasomal degradation in actinobacteria has remained elusive.

In this study, we answer important mechanistic questions regarding the ATP- and pupylation-independent proteasomal degradation in mycobacteria. We investigate the determinants for substrate recruitment by generating unstructured model substrates and testing their stability against Bpa-mediated proteasomal degradation. We also examine the recruitment of the natural substrate HspR by creating substrate variants to identify HspR regions that are important for binding to Bpa and for engagement into the proteasomal degradation chamber. Our findings indicate that although disorder is required, it is not sufficient for substrate selection in Bpa-mediated proteasomal degradation. Furthermore, we show that recruitment of HspR is specific both in terms of substrate binding to the ring-shaped activator and the subsequent threading into the proteolytic chamber of the 20S CP. Expanding our understanding of Bpa's role in protein quality control during bacterial stress responses, we demonstrate that Bpa not only plays a critical role during heat stress in *Mycobacterium smegmatis* but also contributes to the organism's survival under conditions of oxidative stress.

# Results

## Heat shock repressor HspR accumulates in a *M. smegmatis bpa* deletion mutant

It was previously reported that the heat shock repressor HspR accumulates in a *bpa* knockout strain in *M. tuberculosis* and is a Bpa-dependent proteasomal degradation substrate in vitro (Jastrab et al, 2015). To explore if this observation could be corroborated in the soil bacterium *M. smegmatis*, we generated a *M. smegmatis Δbpa* deletion mutant and performed a proteomic analysis comparing the wild-type *M. smegmatis* strain with the *bpa* knockout strain. Each strain was cultivated in biological triplicates under both standard and heat shock conditions. The bacterial proteomes were analyzed using a label-free, quantitative proteomic approach based on data-independent acquisition mass spectrometry. When assessing the relative abundance of HspR, we observed that under heat shock conditions, HspR showed a fivefold increase in abundance in the *bpa* knockout strain compared with the wild-type strain. In fact, even under standard conditions, HspR still accumulated threefold in the knockout over the wild-type strain (Fig 1A and B). This suggests that Bpa-dependent proteasomal degradation controls the levels of HspR in *M. smegmatis*, both under standard and heat shock conditions.

In addition to HspR, we identified other proteins that accumulated in the *bpa* knockout strain, both under standard conditions and during heat shock, under the chosen criteria (>twofold abundance increase, <0.05 adjusted *P*-value). However, among the 22 proteins identified under standard conditions and the 13 proteins identified during heat shock, HspR is the only protein identified for both (Tables S1 and S2). Furthermore, under heat shock, HspR stands out as the most significant hit with the highest fold-change. It can be concluded that HspR is the main Bpa-dependent substrate under heat shock conditions also in *M. smegmatis*.

## Deletion of *bpa* in *M. smegmatis* causes sensitivity to heat shock and oxidative stress during growth in liquid culture

Based on our observation that HspR accumulated significantly in a heat-shocked *M. smegmatis Δbpa* mutant strain, we aimed to determine if the *bpa* knockout strain experienced any growth defects under this stress. We conducted growth experiments with wild type, *bpa* knockout, and *bpa* complemented *M. smegmatis* strains in minimal medium at 37°C or at 45°C. To induce heat shock, the cultures were initially grown at 37°C until reaching the logarithmic phase, diluted, and then subjected to a temperature shift to 45°C. Under standard conditions, the knockout strain displayed no growth impairment. However, when exposed to heat shock conditions, the *bpa* knockout strain exhibited significant growth impairment compared with both the wild type and *bpa* complemented strains. This resulted in a roughly twofold increase in the generation time for the *bpa* knockout strain (9.4 h) compared with either the wild type (4.9 h) or the *bpa*-complemented strain (4.5 h) (Fig 1C).

In addition, aiming to explore another form of stress that could impact protein structure and function, we investigated the effects of oxidative stress on the *bpa* deletion strain in comparison to the wild-type strain by adding 5 mM $H_2O_2$ to cultures grown in minimal

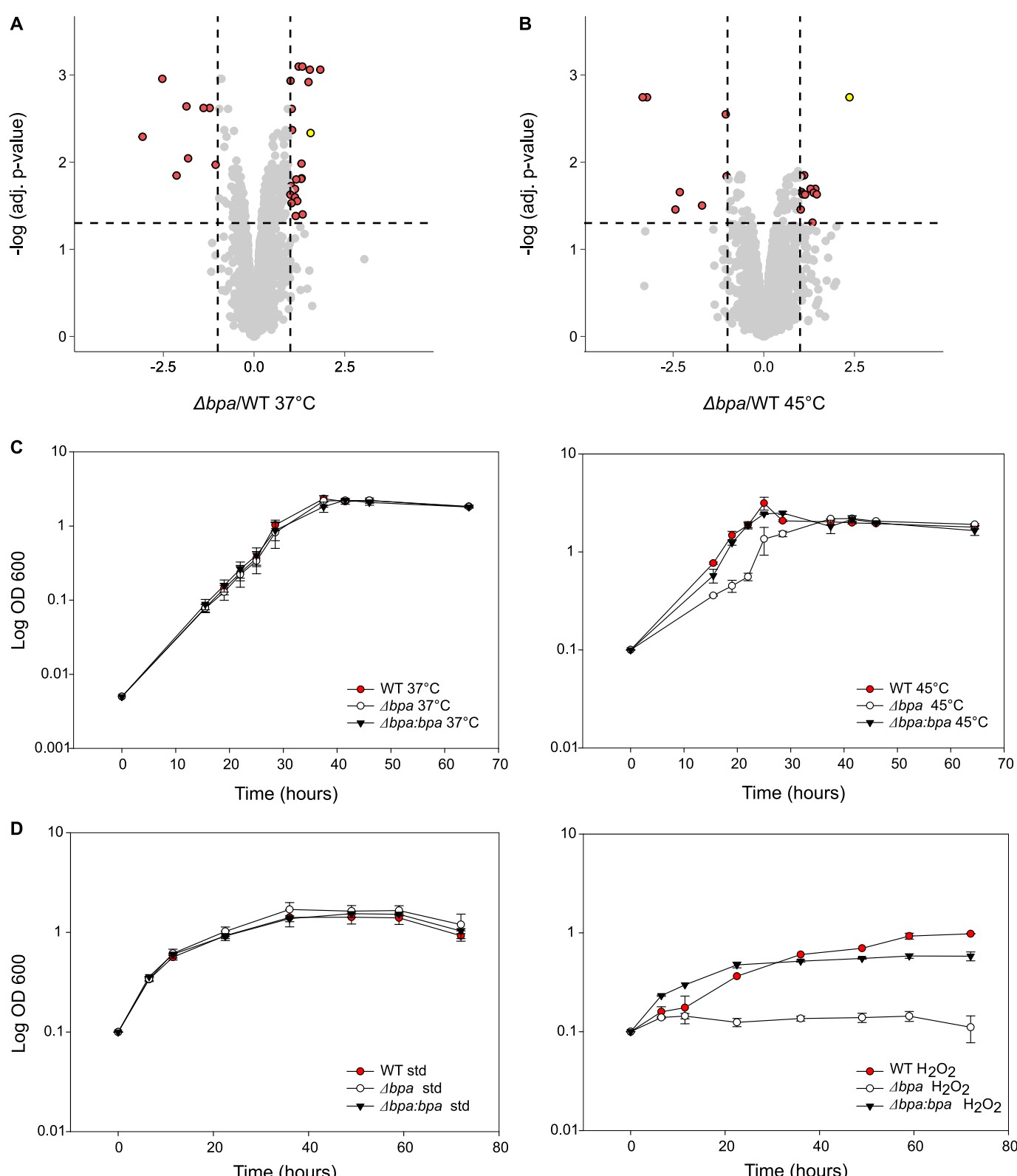

**Figure 1.   *M. smegmatis* Δ*bpa* deletion mutants accumulate heat shock protein HspR and exhibit growth phenotypes under stress.**
**(A)** Volcano plot of protein abundance changes in a Δ*bpa M. smegmatis* deletion strain compared with the wild-type strain when grown under standard conditions at 37°C. Proteins exhibiting significant abundance changes are colored in red. HspR is colored in yellow. Targets pass a cutoff for significance when the adjusted *P*-value is equal to or smaller than 0.05 and the fold change equal to or larger than 2. Strains were grown in biological triplicates. **(B)** Volcano plot of protein abundance changes in a Δ*bpa M. smegmatis* deletion strain compared with the wild-type strain when grown under heat stress at 45°C. Proteins that show significant abundance changes are

medium. Similar to the previous experiment, no growth defect was observed for the *bpa* knockout strain under standard conditions. However, under oxidative stress, the *bpa* knockout strain was no longer viable. This phenotype could be reversed by complementing the knockout strain with *bpa* under its native promoter (Fig 1D).

Taken together, the growth experiments suggest that the *M. smegmatis* mutant strain deficient in *bpa* is impaired under stress conditions known to impact protein structure and function.

### HspR is a Bpa-specific proteasomal degradation substrate in vitro

Based on our proteomics analysis, we identified HspR as the primary Bpa-dependent substrate under heat shock conditions. Next, we explored degradation of HspR in vitro. Bpa lacks an ATPase motor domain that unfolds substrates for translocation into the proteasomal degradation chamber. We therefore hypothesized that Bpa-dependent proteasomal degradation requires inherent instability or disorder in the recruited substrates. Previous studies have shown that mycobacterial HspR undergoes partial unfolding during heat shock, leading to its dissociation from the natural operator and allowing for the transcription of vital heat shock chaperones such as DnaK (Bandyopadhyay et al, 2012). We postulated that DNA-bound HspR would be more stable against proteasomal degradation. To test this hypothesis, we conducted in vitro experiments using recombinantly produced proteins from *M. tuberculosis*. We measured the degradation of HspR in the presence or absence of the HAIR motif DNA. We found that in the absence of DNA, Bpa mediated the rapid turnover of HspR by the 20S CP, whereas the 20S CP alone was unable to degrade HspR (Fig 2A). However, when HspR was incubated with the HAIR motif before the assay, it exhibited significant stabilization against Bpa-mediated proteasomal degradation. These results support our hypothesis that DNA-bound HspR is protected from proteasomal degradation.

Previous studies have also demonstrated that DnaK binds to HspR as a co-repressor to repress its own transcription and the transcription of the other members of the *dnaKJgrpE-hspR* operon under standard conditions (Bucca et al, 2000; von Rosen et al, 2021). Therefore, we investigated whether DnaK could stabilize HspR against degradation by the Bpa–CP complex. In this experiment, we pre-incubated HspR with DnaK before the assay and observed no degradation of HspR during the recorded eight-hour time course. This contrasts with HspR alone, which was completely degraded within ~1 h. These findings suggest that HspR, when in a stable ternary complex with its cognate DNA or the co-repressor DnaK, is either unable to bind to Bpa or cannot be engaged by the proteasome core. Furthermore, by monitoring the temperature-related intensity changes of fluorescently labeled Bpa in a capillary-based titration experiment with HspR, we demonstrated that when in a 1:1 complex

with DnaK or its cognate DNA, HspR no longer interacts with Bpa (Fig S1A). Notably, the stabilization of HspR against Bpa-mediated proteasomal degradation by its cognate DNA is temperature-dependent, as observed in gel-based degradation assays (Fig S1B). At 30°C, HspR in complex with its cognate DNA is completely stabilized against Bpa-mediated proteasomal degradation. At 37°C, less than 50% of HspR is degraded after 8 h. At 42°C (heat shock temperatures), degradation of HspR by the Bpa–CP complex occurs even faster (Fig S1C), albeit still slower than in the complete absence of DNA at 37°C (Fig 2A).

To determine whether HspR is specifically targeted for proteasomal degradation by Bpa, we investigated whether the alternative proteasome interactor Mpa could facilitate the degradation of HspR (Fig 2B). Thus far, attempts to achieve in vitro binding of Mpa to the wild type 20S CP have been unsuccessful, necessitating the use of an "open-gate" variant of the 20S CP (20S CP$^{OG}$), lacking the first seven amino acid residues of the proteasomal α-subunits. By assembling Bpa or Mpa (in the presence of ATP) with the 20S CP$^{OG}$, fully saturating the proteasome to exclude unstructured substrates from entering the degradation chamber directly, we found that the Bpa–CP$^{OG}$ complex rapidly degraded HspR. Under the chosen reaction conditions, the turnover of the substrate was nearly completed within the first 10 min of the reaction time course. In contrast, the Mpa–CP$^{OG}$ complex was unable to degrade HspR, and the substrate remained stable against degradation over the entire eight-hour time course. Therefore, our results indicate that HspR is a Bpa-specific proteasomal degradation substrate. This finding is consistent with the observation that HspR levels increase in a *M. tuberculosis bpa* knockout strain but not in an *mpa* knockout strain (Jastrab et al, 2015).

### HspR is conserved in actinobacteria

To explore the substrate determinants influencing turnover by the Bpa–proteasome complex, we first examined the conservation of HspR domains, which could potentially contribute to substrate recruitment and processing.

We conducted a multiple sequence alignment of HspR orthologs from 20 actinobacterial species using ClustalW (Fig 3A). The alignment showed high conservation of HspR in the DNA-binding helix-turn-helix (HTH) domain and in a short sequence stretch of ~9 residues near the C-terminus (Fig 3A). Using AlphaFold, we then generated a structure prediction for a dimer of *M. tuberculosis* HspR, which predicted regions of disorder in the N-terminal region (1–15) and the C-terminal region (residues 116–125, red) (Fig 3B). The AlphaFold model indicated an α-helical fold for the conserved HTH domain (depicted in blue) and the less conserved C-terminal domain (C-dom) (Fig 3B). Based on the multiple sequence alignment, the AlphaFold model, and disorder predictions (Walsh et al, 2014), we constructed a domain organization model of HspR

---

colored in red. HspR is colored in yellow. Targets pass a cutoff for significance when the adjusted *P*-value is equal to or smaller than 0.05 and the fold change equal to or larger than 2. **(C)** Growth curves of wild-type *M. smegmatis* (WT), Δ*bpa* knockout strain (Δ*bpa*), and complemented strain (Δ*bpa:bpa*) at 37°C (left) and under heat shock at 45°C (right) recorded by optical density measurements. Strains were grown in biological triplicates, and error bars denote the SD. **(D)** Growth curves of wild-type *M. smegmatis* (WT), Δ*bpa* knockout strain (Δ*bpa*), and the complemented strain (Δ*bpa:bpa*) under standard conditions (left) or in the presence of 5 mM H$_2$O$_2$ (right) were recorded by optical density measurements. Strains were grown in biological triplicates, and error bars denote the SD.

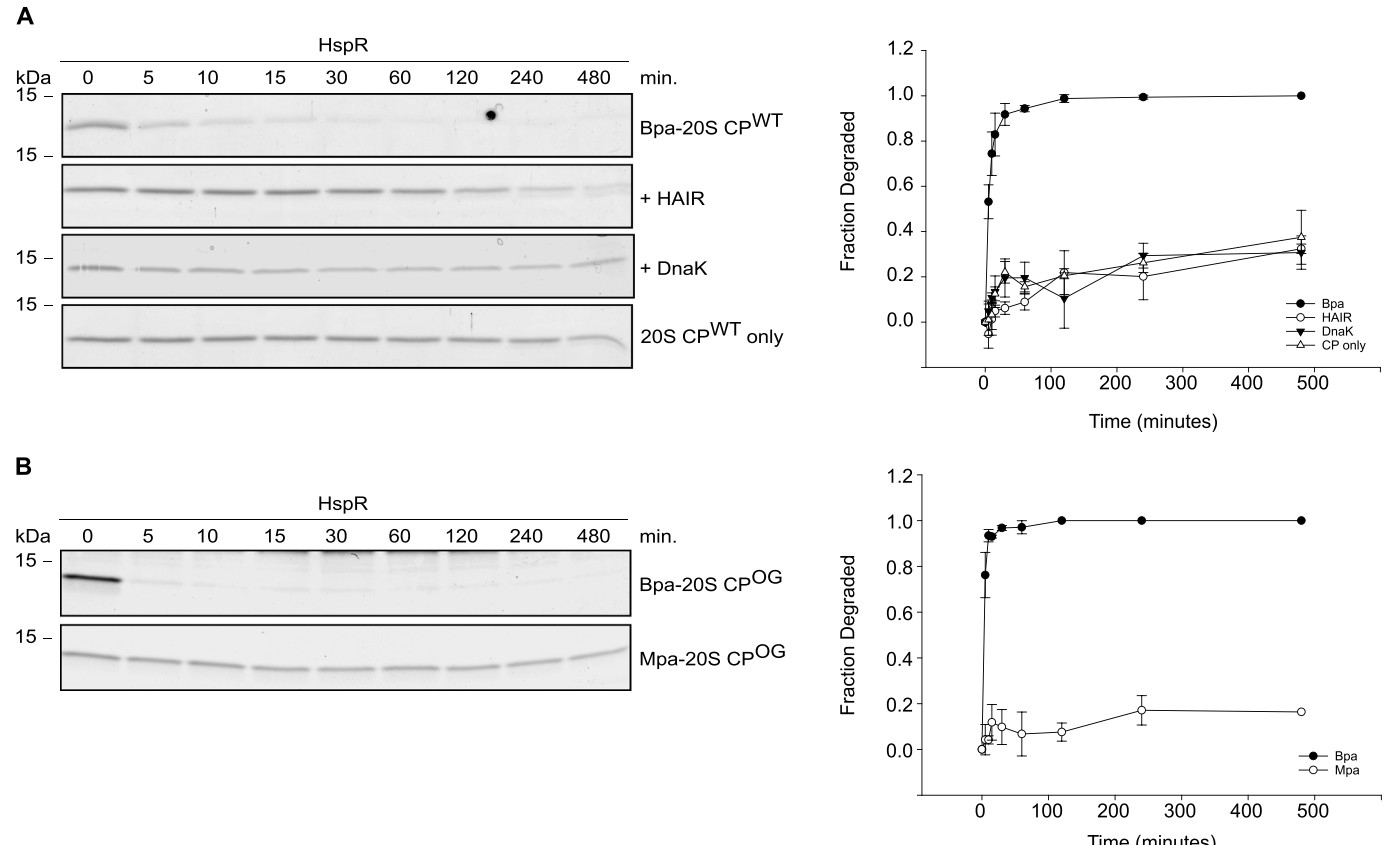

**Figure 2. Proteasomal degradation of heat shock repressor HspR from *M. tuberculosis*.**
**(A)** Proteasomal degradation time courses of natural substrate HspR followed by Coomassie-stained SDS–PAGE. Progress of degradation is measured by following the decrease in HspR gel band intensity (left). In presence of the HAIR operator DNA or in presence of DnaK, HspR is stabilized against Bpa-mediated proteasomal degradation. Decrease in HspR gel band intensity from three independent measurements was quantified by densitometry, with error bars denoting SD between gel band intensities (right). **(B)** Degradation time courses of HspR in presence of Bpa (upper) or in presence of Mpa (lower) and the open-gate proteasome variant (CP^{OG}). Rapid degradation is observed in presence of Bpa but not in presence of Mpa, where HspR remained stable over the entire eight-hour time course (left). Decrease in HspR gel band intensity from three independent measurements was quantified by densitometry, with error bars denoting SD between gel band intensities (right).

(Fig 3C), where the N- and C-terminal sequence regions are predicted to be disordered.

### The conserved disordered C-terminal region of HspR is important for Bpa-mediated degradation

To investigate if the predicted disorder in the N- and C-terminal sequence regions serves as the primary determinant for Bpa-mediated proteasomal degradation, we generated truncated variants of *M. tuberculosis* HspR. These variants lacked either the non-conserved N-terminal stretch (HspRΔN15) or the conserved C-terminal region (HspRΔC9) (Fig 4A). We then recorded Bpa-mediated proteasomal degradation time courses for wild-type HspR (HspR WT) and the two HspR truncation variants (Fig 4B). We found that Bpa in conjunction with 20S CP^{WT} rapidly degraded HspR WT, achieving nearly complete turnover within 5 min under the chosen conditions. In contrast, the 20S CP^{WT} alone was unable to degrade HspR within an eight-hour time course under the same conditions (Fig 4B, top). When a hexapeptide carrying the GQYL proteasomal interaction motif, capable of inducing gate-opening by inserting itself between the

α-subunits of the 20S CP, was used at sufficiently high concentration (200 μM), it supported proteasomal degradation of HspR WT almost as effectively as Bpa, resulting in near-complete substrate turnover after 30 min. Similarly, the permanently open proteasome variant 20S CP^{OG}, where the α-gate was truncated, efficiently turned over HspR within 15 min under the chosen conditions.

The Bpa-dependent proteasomal degradation of the C-terminally truncated HspR variant, HspRΔC9, occurred significantly slower compared to degradation of HspR WT. Complete degradation was achieved only after about 2 h under the chosen conditions (Fig 4B, middle). When the hexapeptide with the C-terminal GQYL motif was used instead of Bpa to induce gate-opening, the reaction was even slower, taking ~8 h to complete. In the absence of Bpa or the hexapeptide, HspRΔC9 remained stable against degradation by 20S CP^{WT} throughout the entire time course. However, the genetically engineered open-gate variant, 20S CP^{OG}, degraded HspRΔC9 within 15 min, only slightly slower than the degradation of HspR WT by 20S CP^{OG}.

Bpa-mediated proteasomal degradation of the N-terminally truncated HspR variant, HspRΔN15, occurred even slightly faster

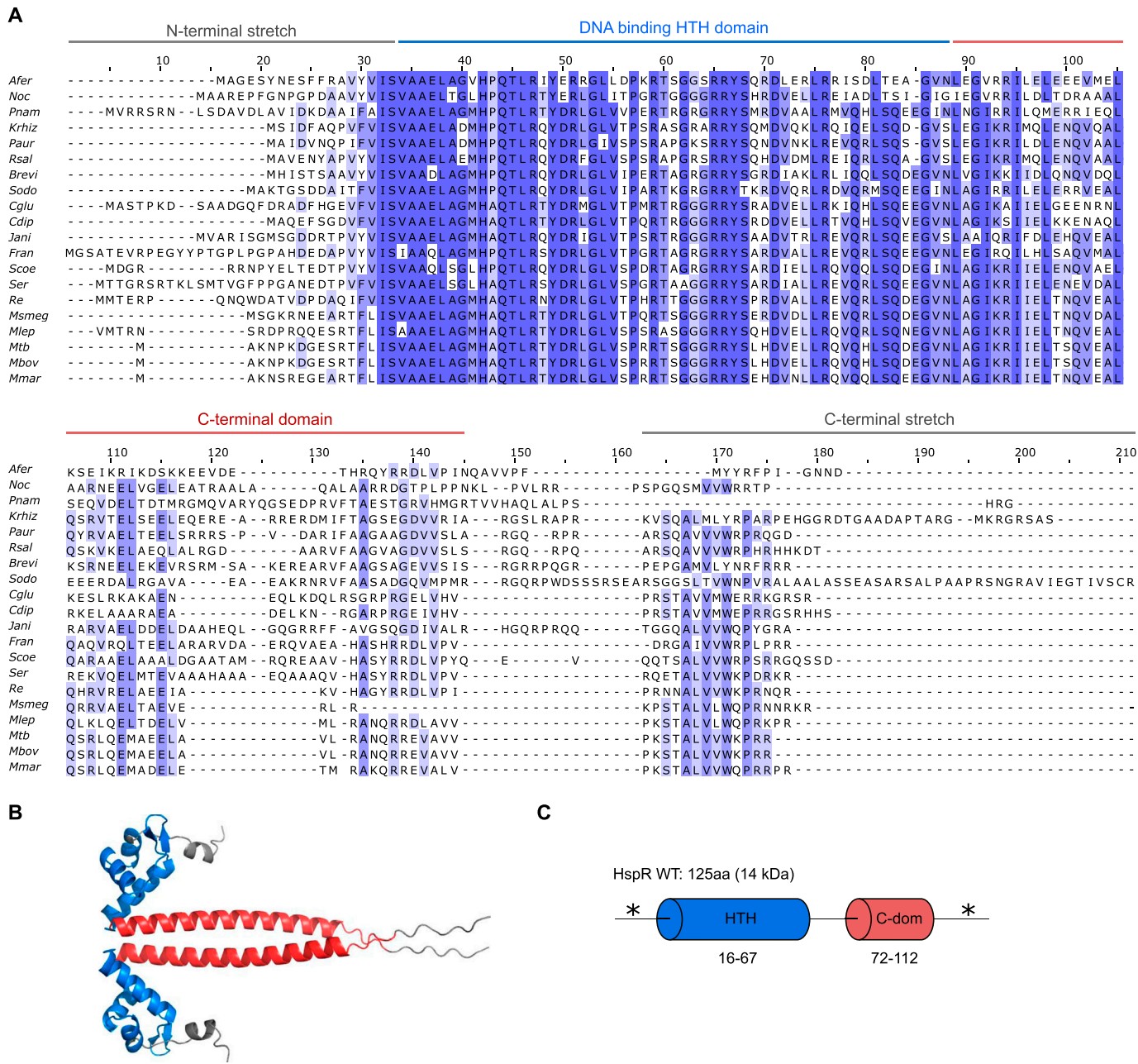

**Figure 3. HspR sequence conservation and domain organization.**
**(A)** Multiple sequence alignment of HspR sequences from various actinobacteria. Amino acid sequences were aligned in ClustalW (Larkin et al, 2007) and viewed in Jalview 2.11 (Waterhouse et al, 2009). Residues with an identity score greater than 0.5 are shaded in blue with color intensity increasing from light (0.5) to dark blue (1). Organisms are abbreviated in the following way: Afer (*Acidithrix ferrooxidans*), Noc (*Nocardioides sp.*), Pnam (*Propionibacterium namnetense*), Krhiz (*Kocuria rhizophila*), Paur (*Paenarthrobacter aurescens*), Rsal (*Renibacterium salmoninarum*), Brevi (*Brevibacterium sp.*), Sodo (*Schaalia odontolytica*), Cglu (*Corynebacterium glutamicum*), Cdip (*Corynebacterium diphtheriae*), Jani (*Janibacter sp.*), Fran (*Frankia alni*), Scoe (*Streptomyces coelicolor*), Ser (*Saccharopolyspora erythraea*), Re (*Rhodococcus erythropolis*), Msmeg (*Mycobacterium smegmatis*), Mtb (*Mycobacterium tuberculosis*), Mbov (*Mycobacterium bovis*), Mmar (*Mycobacterium marinum*). **(B)** Structural model of an HspR^Mtb dimer. The structural model was generated in AlphaFold (Jumper et al, 2021; Varadi et al, 2022). The N-terminal region containing the DNA-binding domain is colored in blue. The C-terminal α-helix is colored in red. **(C)** Domain organization of HspR from *M. tuberculosis*. The DNA-binding helix-turn-helix (HTH) domain is colored in blue. The C-terminal α-helix domain (C-dom) is colored in red. The predicted regions of disorder are indicated (*) as determined by the PASTA 2.0 prediction tool (Walsh et al, 2014).

than the degradation of HspR WT. This indicates that the N-terminal disordered region does not facilitate Bpa-mediated proteasomal degradation (Fig 4B, bottom). Substituting Bpa with the hexapeptide

containing the GQYL interaction motif, which induces gate-opening of the 20S CP^WT, led to slower degradation of the HspRΔN15 variant, resulting in complete degradation of the substrate within 30 min

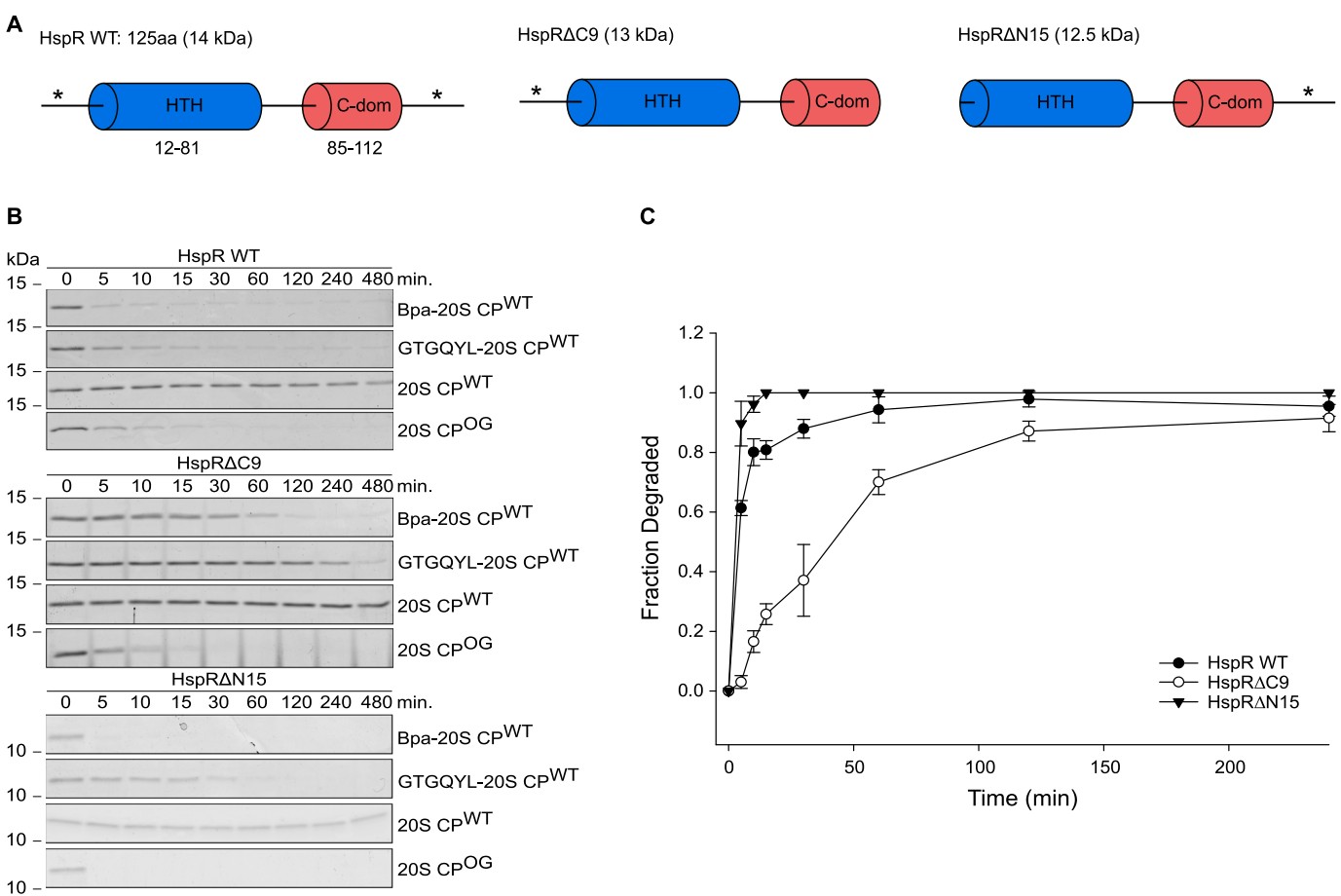

**Figure 4. Bpa-mediated proteasomal degradation of HspR truncation variants.**
**(A)** Domain organization of HspR variants. Wild type HspR (HspR WT) consists of 125 amino acid residues, featuring an N-terminal DNA-binding HTH domain (blue) and a C-terminal domain (C-dom, red) with predicted disorder (denoted by *) in the N-terminal (15 residues) and C-terminal (nine residues) regions. The C-terminal truncation variant (HspRΔC9) lacks the unstructured C-terminal stretch, whereas the N-terminal truncation variant (HspRΔN15) lacks the unstructured N-terminal stretch preceding the HTH domain. **(B)** Proteasomal degradation time courses of HspR WT, HspRΔC9, and HspRΔN15 followed by Coomassie-stained SDS–PAGE. Degradation time courses are carried out in the absence or presence of Bpa or in the presence of a hexapeptide carrying the proteasome interaction motif (GTGQYL-20S CP[WT]) for gate opening. **(C)** Decrease in HspR gel band intensity for HspR WT and the truncation variants from three independent measurements of degradation by Bpa-20S CP[WT] was quantified densitometrically, with error bars denoting SD between gel band intensities. Representative time courses are shown in (B).

compared with 5 min required for the Bpa-mediated reaction. In the absence of gate-opening, HspRΔN15 remained stable over the course of 8 h, whereas the engineered open-gate variant degraded HspRΔN15 within 5 min. The degradation behavior of the HspRΔN15 variant was thus comparable to what we observed for HspR WT (Fig 4C). Collectively, these results suggest that the disordered C-terminal region of HspR, but not its N-terminal region, acts as a degradation determinant for Bpa-mediated proteasomal degradation. However, the role of the C-terminal region in substrate binding or threading remained unclear at this stage of the study.

### Disorder is not sufficient for Bpa-mediated proteasomal degradation

To further explore the role of disorder in Bpa-mediated proteasomal degradation, we examined two model substrates in their native and permanently non-native conformations (Fig S2A and B). Were conformational state of a protein to be the sole determinant

for substrate recruitment, the non-native form of the model substrate should be readily recruited and degraded by the Bpa–CP complex, whereas the folded form of the substrate should remain stable against Bpa-mediated proteasomal degradation. One of the model substrates used was ribonuclease A (RNaseA) from the bovine pancreas. Native RNaseA is stabilized by four disulfide bridges between eight cysteine residues. To generate a non-native form of this model substrate, we unfolded RNaseA under reducing conditions and modified the cysteine residues by S-carboamido-methylation using iodoacetamide. Intact mass spectrometry confirmed the complete alkylation of all eight cysteine residues (Fig S2C). The secondary structure composition of folded and unfolded RNaseA was assessed by circular dichroism, with folded RNaseA exhibiting expected contributions of both α-helical and β-sheet secondary structure elements, which were absent in the carba-midomethylated RNaseA preparation (Fig 5A, left). The second conformational model substrate was FimA, the main subunit of the type 1 pilus rod from uropathogenic *Escherichia coli*. Protomers in

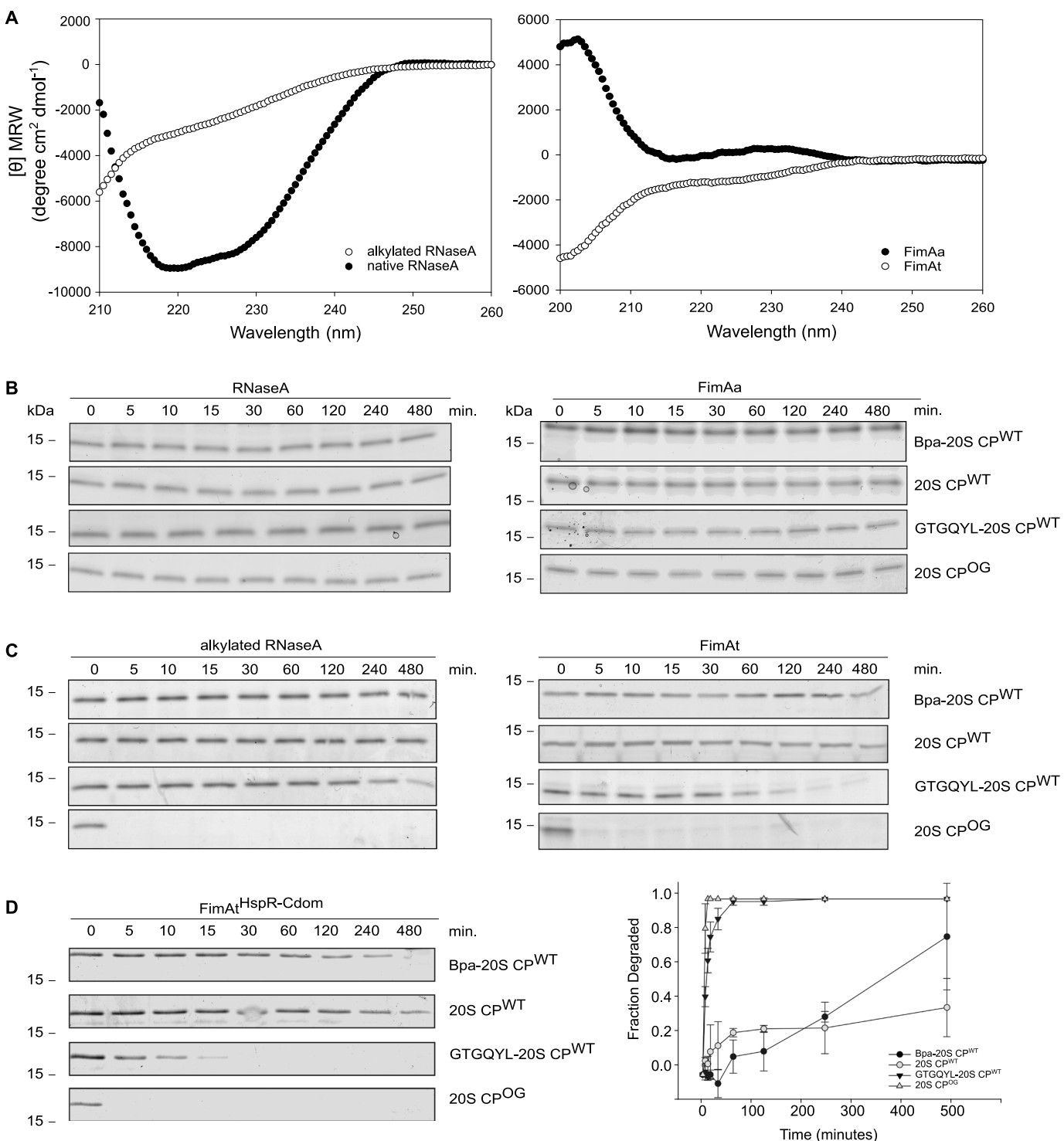

**Figure 5. Proteasomal degradation assays with conformational model substrates.**
**(A)** CD spectra of folded RNaseA and alkylated RNaseA (left) and CD spectra of the hyperstable FimAa variant and the unstable FimAt variant (right). CD spectra are an average of five acquisitions and are buffer corrected. **(B, C)** Proteasomal degradation time courses of different conformational states of two model substrates followed by Coomassie-stained SDS–PAGE. **(B)** Degradation time courses were recorded of folded forms of RNaseA (left) and FimAa (right) and **(C)** of the respective non-native forms (left and right gels, respectively). Degradation assays were performed with wild-type 20S proteasome in presence or absence of Bpa, or in presence of hexapeptide carrying the proteasomal interaction motif (GTGQYL). As a positive control, proteins were subjected to degradation by the open-gate proteasome variant (20S CP$^{OG}$). **(D)** Proteasomal degradation assays were recorded of the unstable model substrates FimAt to which the C-terminal domain of HspR was fused (FimAt$^{HspR-Cdom}$). Degradations were carried out with wild-type 20S proteasome in presence or absence of Bpa, or with hexapeptide carrying the proteasomal interaction motif (GTGQYL). As a positive control, FimAt$^{HspR-Cdom}$ was subjected to degradation by the open-gate proteasome variant (20S CP$^{OG}$) (left).

the assembled pilus engage in a domain-swap interaction known as donor-strand complementation. FimA possesses an incomplete immunoglobulin-like fold lacking the C-terminal β-strand, which is supplied by the subsequent pilus subunit carrying the missing strand as an N-terminal extension (the donor strand) (Puorger et al, 2011). We used a variant called FimAa, where the missing N-terminal donor-strand is fused to the C-terminus, resulting in a self-complemented, highly stable monomer. Within the time frame of our experiments, FimAa can be considered permanently folded, as unfolding occurs on average only once every $10^{12}$ yr. Another variant, FimAt, lacks the N-terminal extension, rendering it permanently unfolded, as indicated by the comparison of the CD spectra of FimAa and FimAt (Fig 5A, right) (Puorger et al, 2011).

To assess the conformational state as a determinant for substrate recruitment, we performed degradation experiments using the folded and non-native forms of the two model substrates. Our degradation time courses show that Bpa cannot facilitate the proteasomal degradation of the folded model substrates (Fig 5B). Both native RNaseA and the permanently folded FimA variant, FimAa, remained completely stable against Bpa-mediated proteasomal degradation (Fig 5B). Likewise, gate-opening with the GQYL-containing hexapeptide did not result in the degradation of native RNaseA or FimAa by the 20S CP. Even the engineered open-gate proteasome variant was unable to degrade either of the two natively folded substrates. These results provide further evidence for the principle of compartmentalization used by the proteasome and related degradation machines, wherein stably folded domains are unable to gain access into the degradation chamber.

In contrast, the engineered open-gate proteasome efficiently degraded the unfolded forms of RNaseA and FimAt, demonstrating that fully disordered proteins can diffuse through the ungated α-ring pore and be degraded. However, remarkably, we observed no degradation of the unfolded form of either model substrate by the Bpa–CP complex (Fig 5C). Substituting Bpa with the GQYL-containing hexapeptide led to slow degradation, with almost all FimAt degraded by the end of the eight-hour time course, and approximately half of the non-native RNaseA degraded within the same time frame. These results suggest that Bpa provides an additional layer of selectivity and that although conformational disorder is necessary for entry into the proteasomal degradation chamber, it alone is insufficient to render a protein a substrate for the Bpa-CP complex.

Having established that the disordered C-terminal stretch of HspR contributes to Bpa-dependent turnover of HspR (Fig 4B), next, we examined if we could render the unfolded FimA variant, FimAt, a substrate for Bpa-mediated proteasomal degradation by creating a C-terminal fusion with the C-terminal domain of HspR (FimAt$^{HspR-Cdom}$). However, the fusion did not significantly enhance the degradation of FimAt by the Bpa–CP complex (Fig 5D). Interestingly, we observed that FimAt$^{HspR-Cdom}$ was degraded by 20S CP$^{WT}$ when the GQYL-containing hexapeptide was used to open the gate, resulting in complete degradation within 30 min. This suggests that the HspR C-terminal disordered region facilitates diffusion into the

proteasome chamber, but additional determinants are required for efficient Bpa-mediated proteasomal degradation.

## The conserved disordered C-terminal stretch of HspR is a specific threading motif for Bpa-mediated proteasomal degradation

Our findings indicate that the C-terminal stretch of HspR plays a significant role in Bpa-mediated proteasomal degradation. For a more quantitative comparison of the degradations of HspR WT and HspRΔC9, we conducted gel densitometric analysis of their initial degradation rates (Fig S3A and B). The comparison showed that HspR WT is degraded an order of magnitude faster than the C-terminal truncation variant. The apparent initial velocity of degradation of wild-type HspR by the Bpa–proteasome complex under the chosen conditions is 0.65 µM/min, whereas that of the C-terminal truncation variant is 0.06 µM/min. However, the specific involvement of the C-terminal disordered region in substrate recruitment remained unclear. One explanation could be that this part of HspR serves as a direct binding determinant. Alternatively, the disordered region could act as a facilitator for threading, as its extended conformation and lack of tertiary interactions would enable diffusion of the C-terminal tail across the open proteasomal gate.

Previous studies suggested hydrophobic rings within the dodecameric core of Bpa are involved in hydrophobic interactions between substrates (Hu et al, 2018). The conserved C-terminal–disordered region of HspR contains several hydrophobic residues that could potentially participate in such interactions (Fig 3A). To investigate the involvement of the C-terminal region in HspR binding to Bpa, we fluorescently labeled Bpa and monitored the temperature-related fluorescence intensity changes in a capillary-based titration experiment with HspR WT and HspRΔC9 (Fig 6A). If the C-terminal region of HspR was critical for Bpa binding, we should observe a higher dissociation constant for HspRΔC9 than HspR WT. However, we found that the normalized fluorescence intensity changes were similar for HspR WT and HspRΔC9, with comparable calculated dissociation constants of 5.60 ± 0.97 µM and 3.62 ± 0.58 µM, respectively. These results suggest that HspR WT and HspRΔC9 bind Bpa with similar affinities, and the conserved C-terminal region of HspR does not significantly contribute to the binding to Bpa.

Although disorder appears to be important for Bpa-dependent substrate degradation, HspR WT possesses a significant amount of α-helical secondary structure, which is maintained in the truncated variant HspRΔC9 (Fig 6B). Previous studies have suggested that the C-terminal sequence stretch of HspR modulates its thermal stability (Bandyopadhyay et al, 2012). Therefore, we measured thermal unfolding transitions of the wild-type HspR and the HspRΔC9 variant (Fig 6C). Indeed, we observed that HspRΔC9 had a higher melting temperature ($T_m$ = 320.9 ± 1.8 K/47.8 ± 1.8°C) than HspR WT ($T_m$ = 314.8 ± 0.6 K/41.6 ± 0.6°C). However, we found that the degradation defect upon deletion of the C-terminal stretch was not temperature dependent (Fig S3C and D). Bpa-mediated

---

Decrease in FimAt$^{HspR-Cdom}$ gel band intensity from three independent measurements was quantified by densitometry, with error bars denoting SD between gel band intensities (right).

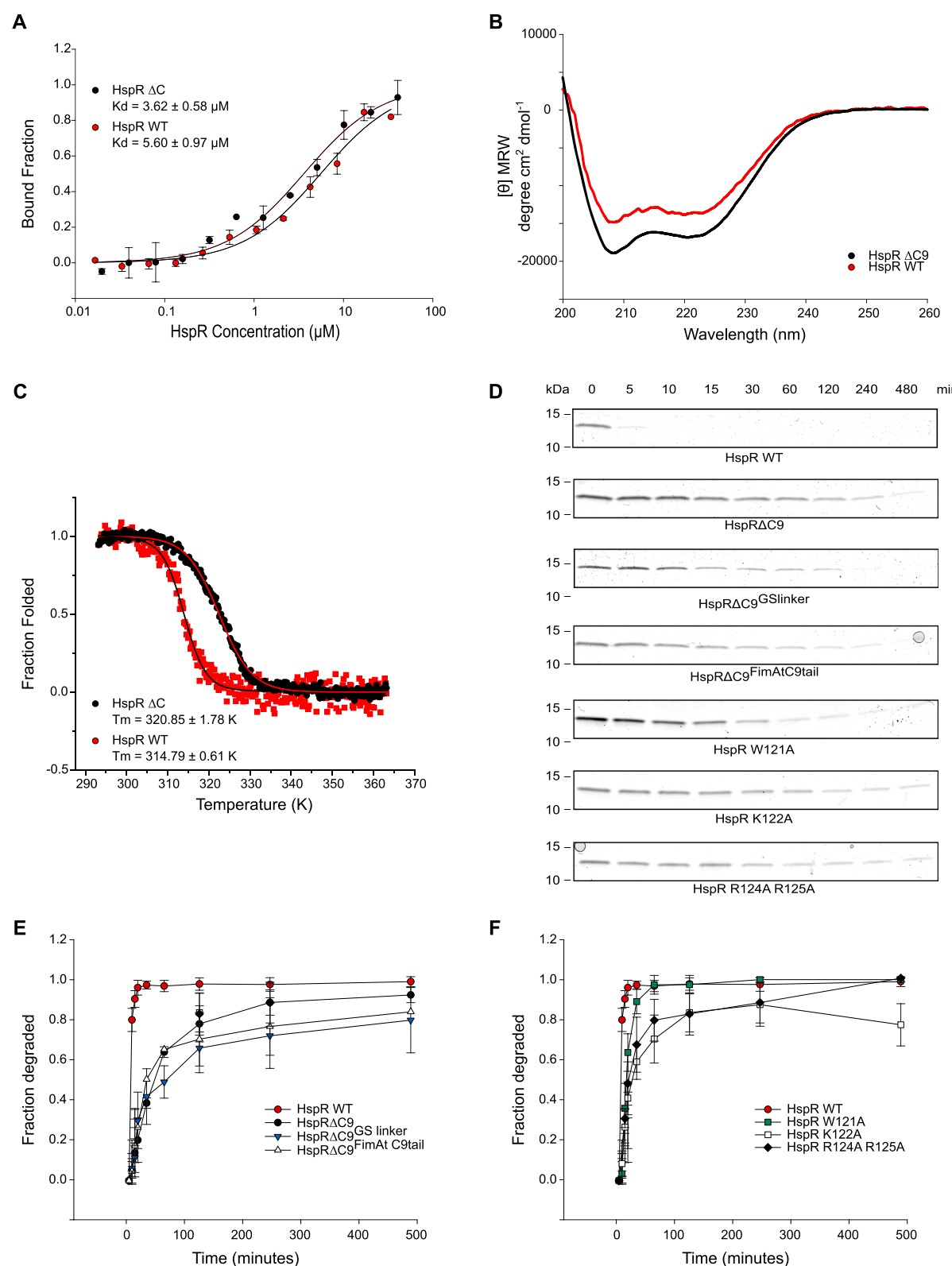

**Figure 6. HspRΔC9 binds Bpa with similar affinity as HspR WT but its proteasomal degradation is impaired.**
**(A)** Binding of HspR WT or HspRΔC9 to fluorescently labeled Bpa was measured by recording the temperature-related fluorescence intensity changes at different concentrations of substrate. Reactions were run in triplicates and dependency of bound fraction of Bpa to substrate was fitted using the company-provided analysis software. Error bars denote SD. **(B)** CD spectra of HspR WT and HspRΔC9. Both substrates display the CD signature of a folded protein with strong α-helical secondary structure contribution. CD spectra are an average of five acquisitions and are buffer corrected. **(C)** Thermal unfolding transitions of HspR WT and HspRΔC9 recorded by

proteasomal degradation of both HspR WT and HspRΔC9 was slower at lower temperature (30°C) compared with 37°C. Yet, the degradation defect of HspRΔC9 was not reverted when conducting the gel-based degradation assay at 42°C. We monitored the decrease in gel band intensity of HspR and HspRΔC9 and analyzed them using densitometry (Fig S3D).

Considering that the conserved, disordered C-terminal stretch of HspR is crucial for efficient protein turnover, we explored whether this degradation defect could be reversed by complementing HspRΔC9 in vitro with a random GS linker (GGS GSS GSG) or the C-terminal sequence region of an unrelated unfolded protein. We fused either the nine-residue GS linker or the last nine residues of the unfolded model substrate FimAt to the C-terminus of HspRΔC9 and tested these variants for Bpa-mediated proteasomal degradation (Fig 6D). However, neither HspRΔC9$^{GSlinker}$ nor HspRΔC9$^{FimAtC9tail}$ rescued the degradation defect of HspRΔC9. In addition to substituting the C-terminal stretch of HspR, we assessed individual residues in this region (W121, K122, R124, and R125) for their contribution to efficient substrate degradation (Fig 6D). Single- or double-point mutations of these residues (W121A,K122A, and R124A R125A) severely impaired HspR degradation. We monitored the decrease in gel band intensity of HspR and analyzed them using densitometry (Fig 6E and F). These results suggest that although the C-terminal region of HspR is not a major binding determinant, it exhibits a sequence preference that contributes to the facilitation of HspR degradation by the Bpa–CP complex. The data support a role for the C-terminal disordered region of HspR as a sequence-dependent threading determinant for Bpa-mediated proteasomal degradation.

## The N-terminal HTH domain of HspR contributes to substrate recruitment for Bpa-mediated proteasomal degradation

Based on our findings indicating a negligible contribution of the C-terminal–disordered region of HspR to the affinity for Bpa, we next investigated which part of HspR is involved in the initial capture complex with Bpa before threading into the 20S CP. We designed peptides spanning the sequence of HspR and assessed the binding of these peptides to fluorescently labeled Bpa. In total, we probed eight 15-amino acid peptides covering residues 16–121 of HspR (Fig 7A and Table S3). Peptides containing either the full or partial epitope of the Bpa-interacting region were expected to exhibit binding, whereas peptides spanning regions of HspR that do not interact with Bpa were anticipated to show no binding. Given that we had previously established that the disordered N-terminal stretch of HspR (1–15) did not contribute to substrate recruitment, we excluded this sequence from our peptide screen.

To evaluate the binding interactions, we fluorescently labeled Bpa and monitored the temperature-related fluorescence intensity changes in a capillary-based titration experiment. The peptides

were titrated against the fluorescently labeled Bpa, covering a concentration of 1 mM–0.49 $\mu$M. Initially, we performed a qualitative screen to identify peptides that induced a signal change (Fig 7B). Among the peptides tested, only peptide 1, which corresponds to residues 16–30 within the HTH domain of HspR, produced significant changes. We then quantified the binding of peptide 1 to Bpa by monitoring the temperature-related fluorescence intensity change. The analysis showed that peptide 1 bound Bpa with a dissociation constant of 0.33 ± 0.09 mM (Fig 7C).

To further probe which residues might be involved in the interaction between HspR and Bpa, we focused on conserved charged amino acids within the region spanning residues 16–30. The entire 15-residue stretch exhibits high conservation across actinobacteria (Fig 7D). Among these conserved residues, Glu19 and His24 were chosen as potential interaction candidates. In addition, R29, which we later demonstrate to be crucial for DNA binding, also was included. Although several hydrophobic residues are also strongly conserved, the AlphaFold structure prediction suggested their primary role in stabilizing the fold of the helix bundle. Consequently, we focused on the charged residues that were predicted to be more exposed and pointing outward, making them more likely to engage with other binding partners. We constructed mutants targeting these residues (HspR H24A, HspR E19K, HspR E19A, and HspR R29A). In case of E19, we included a mutation to a residue with opposite charge in addition to the alanine mutation.

Subjecting the HspR HTH variants of residues E19 and H24 to Bpa-mediated proteasomal degradation, we observed that mutation of H24 or E19 resulted in a notable degradation defect (Fig 7E). Unlike HspR WT, which was degraded within 5 min, it took 2 h to degraded about 90% of the HspR H24A variant. The glutamate variants exhibited an even slower degradation, without reaching complete degradation even after 8 h. To ensure that the observed defect was not due to a loss of structure, we recorded CD spectra of the HspR H24A, HspR E19A, and HspR E19K variants (Fig S4A–D), and found that they, like the wild-type protein, displayed a CD signature with the characteristic strong $\alpha$-helical contribution. Although Bpa-mediated degradation was impaired, as expected, these HspR HTH variants could still be degraded by the hyperactive "open gate" proteasome (Fig S4E). These results indicate that residues in the N-terminal HTH domain of HspR play a role in recruiting the protein for Bpa-mediated proteasomal degradation through direct interaction with Bpa. Furthermore, temperature-related changes in fluorescently labeled Bpa in the capillary-based titration experiment showed that the HspR H24A variant bound Bpa less tightly than HspR WT, exhibiting a dissociation constant of 12.06 ± 4.71 $\mu$M compared with 3.56 ± 0.62 $\mu$M observed for the wild type (Fig S4F).

In contrast to the mutations in H24 or E19, the HspR R29A variant displayed no defect in Bpa-mediated proteasomal degradation, indicating that this positively charged residue does not participate

---

following the CD signal at 222 nm. Transitions were recorded in triplicates and one representative transition is shown. **(D)** Bpa-mediated proteasomal degradation time courses of HspR WT, HspRΔC9, HspRΔC9 fused with a random GS linker of nine residues (HspRΔC9$^{GSlinker}$), HspRΔC9 fused to the last nine residues of model substrate FimAt (HspRΔC9$^{FimAtC9tail}$), HspR HTH variants HspR W122A, HspR K122A, HspR R124A R125A. Degradation was followed by Coomassie-stained SDS–PAGE. **(E, F)** Densitometric analysis of gel bands shown in (D), taken from the gel triplicates, with error bars denoting SD between gel band intensities. Time course for HspR WT is shown twice, once in each graph to allow direct visual comparison of all traces to HspR WT trace.

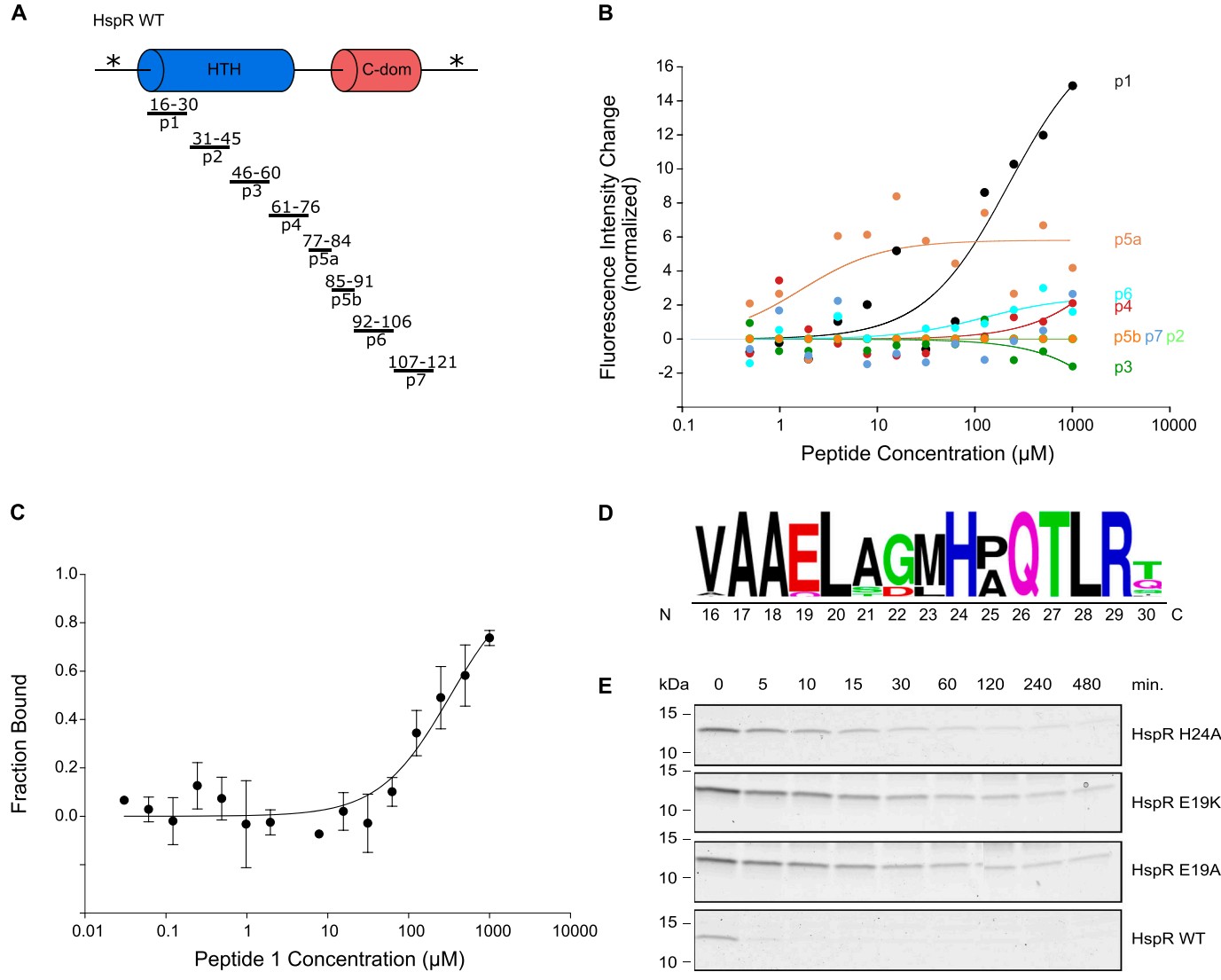

**Figure 7. HspR peptide–binding array.**
**(A)** Overview of peptides used in the peptide-binding array covering the amino acid sequence of HspR. **(B)** Qualitative assessment of signal change in the peptide-binding assay of peptides 1–7 to fluorescently labeled Bpa, using temperature-related fluorescence intensity changes. Dependency of signal change on substrate concentration was fitted using the company-provided analysis software. **(C)** Binding of peptide 1 to fluorescently labeled Bpa, measured by temperature-related fluorescence intensity changes. Reactions were run in triplicates and dependency of bound fraction of Bpa was fitted using the company-provided analysis software. Error bars denote SD. **(D)** Logo of conservation of the N-terminal region of the HTH domain as assessed by multiple sequence alignment from Fig 3A. Relative conservation is depicted by increasing size of residue letter. **(E)** Bpa-mediated proteasomal degradation time courses of HspR HTH domain variants were followed by Coomassie-stained SDS–PAGE.

in the recruitment process (Fig S5A). Interestingly, although the wild-type HspR is protected from Bpa-mediated proteasomal degradation in the presence of its cognate DNA (Figs 2A and S5A), the HTH variant R29A was still degraded by the Bpa–CP complex in the presence of DNA containing the HAIR motif (Fig S5A). We quantified the decrease in gel band intensity of HspR using densitometry (Fig S5B). Our results suggest that residue R29 is involved in the binding of HspR to the HAIR motif. Notably, the stabilization of HspR through binding to cognate DNA with the HAIR motif even confers protection against degradation by the hyperactive open-gate proteasome, 20S $CP^{OG}$. Our results suggest that an effect on overall conformational stability is responsible,

rather than a local effect on the C-terminal–threading region because the experiment was conducted with the C-terminally truncated variant, HspRΔC9, which is efficiently degraded by the open-gate proteasome in absence of HAIR DNA but is rendered degradation-resistant in its presence (Figs S5C and 4B, middle gel).

## The HspR co-repressor DnaK is a pupylation and proteasomal degradation substrate

Previous proteomic studies in mycobacteria have suggested that DnaK, a heat shock chaperone and the co-repressor of HspR,

may be a target for pupylation (Festa et al, 2010; Poulsen et al, 2010). This implies a potential interplay between pupylation-dependent Mpa-mediated proteasomal degradation, ATP-independent Bpa-mediated proteasomal degradation, and the *dnaKJgrpE-hspR* operon. To demonstrate the pupylation of DnaK in vitro, we conducted pupylation assays (Fig S6A). In the presence of the PafA ligase and ATP, we observed near-complete pupylation of DnaK after 20 h. Pupylation of DnaK is less efficient compared to pupylation of the well-established pupylation substrate PanB. The presence of additional protein bands above 70 kD indicates the pupylation of multiple lysine residues in DnaK. Mass spectrometric analysis of DnaK–Pup identified pupylated lysine residues at K159, K469, K483, and K522 (source data Fig S6). Notably, K159, K469, and K483 are located in the nucleotide-binding domain, whereas K522 is found in the substrate binding domain. Next, we tested whether pupylated DnaK could serve as a proteasomal degradation substrate in vitro. We found that the Mpa–CP$^{OG}$ complex could degrade DnaK–Pup in vitro, whereas the open-gate proteasome alone could not (Fig S6B). These findings provide experimental evidence that DnaK is a pupylation and proteasomal degradation substrate.

# Discussion

Proteasomal degradation in mycobacteria and other actinobacteria follows a modular principle involving at least two alternative proteasome activators, Mpa and Bpa, which mediate distinct degradation pathways. The Mpa-mediated pathway is well-understood and relies on the post-translational modification of stable protein substrates with Pup, followed by ATP-dependent degradation by the Mpa–proteasome complex. In contrast, Bpa does not recognize pupylated proteins and lacks ATPase activity, suggesting an alternative mechanism for substrate degradation. However, the nature of this mechanism has remained unclear.

In our study, we investigated the conformational hypothesis of substrates recruitment and found that partial disorder supports substrate processing but not substrate recruitment. In fact, our results support a role of Bpa as a selective filter, preventing degradation of substrates solely on conformational principles. Using HspR as an example, a known Bpa-dependent proteasomal substrate (Jastrab et al, 2015), we show that Bpa engages in specific interactions with the substrate before the disordered C-terminal region of HspR threads across the proteasomal gate. Our analysis suggests that Bpa-dependent proteasomal degradation involves a combination of conformational selection and substrate-specific interactions.

Previous hypotheses suggested that Bpa facilitates the degradation of unstable substrates and recruits denatured/damaged proteins to the proteasome during stress. These hypotheses were based on observations that Bpa contributes to survival during heat stress in *M. tuberculosis* (Jastrab et al, 2015, 2017) and facilitates proteasomal degradation of β-casein and partially disordered HspR (Delley et al, 2014; Jastrab et al, 2015; Bolten et al, 2016). Parallels can also be drawn to ATP-independent proteasomal degradation in eukaryotes (Huang et al, 2016), where the activator PA200 aids in the degradation of unstable substrates like the

amyloidogenic protein tau. However, the mechanisms of substrate recruitment by ATP-independent activators in eukaryotes are not fully understood.

Our study demonstrates that a disordered region is necessary for Bpa-mediated proteasomal degradation and serves as a degradation determinant. Deleting the C-terminal nine amino acids of HspR leads to a significant degradation defect (Figs 4B and C and S3), highlighting the importance of the disordered C-terminal region for efficient degradation. Interestingly, when HspR forms a complex with its co-repressor, DnaK, it becomes protected against Bpa-mediated degradation (Fig 2). DnaK binds to the C-terminal region of HspR, masking the degradation determinant and potentially hindering binding of HspR to the Bpa funnel (Fig S1). These findings parallel the requirements for unstructured regions on substrates in Mpa-mediated, Pup-dependent degradation and eukaryotic, ubiquitin-dependent proteasomal degradation. Pup not only provides a binding determinant but also an unstructured region that enables threading into the Mpa central pore (Sutter et al, 2009; Burns et al, 2010; Striebel et al, 2010; Wang et al, 2010). Similarly, the eukaryotic 26S proteasome requires a disordered threading handle in addition to polyubiquitination to degrade substrates (Prakash et al, 2004).

Notably, the disordered C-terminal region on HspR appears to have some sequence specificity, as replacing it with random sequences or sequences from another protein results in inefficient degradation. In the eukaryotic 26S proteasome, threading efficiency correlates with the "complexity" of the amino acid sequence of the disordered stretch, which enhances engagement with the ATPase AAA motor. However, in the Bpa–proteasome complex, engagement of substrates is likely driven by different requirements, as the disordered region does not interact with an AAA motor spiral staircase, but rather inserts directly into the proteasomal α-ring gate. Prevention of back-sliding is likely governed by binding of the unstructured region to the inner walls of the 20S chamber, with regular sampling of the unfolded state enabling further diffusion into the degradation chamber. The overall stability is also likely to affect the processive nature of degradation, as the presence of the C-terminal tail on HspR WT lowers its melting temperature and potentially promotes directional diffusion into the degradation chamber, especially during heat shock.

Although an unstructured region is a crucial determinant for Bpa-mediated degradation, disorder alone is not sufficient, as Bpa is unable to facilitate proteasomal degradation of unfolded model substrates (Fig 5). Furthermore, truncation of the disordered C-terminal stretch of HspR, which is required for efficient proteasomal degradation, does not eliminate binding of HspR to Bpa (Fig 6). These findings suggest that Bpa recruits HspR for degradation through a stepwise process, involving the formation of a binary complex between HspR and Bpa via specific recognition sites, followed by engagement of the disordered C-terminal tail into the proteasomal α-ring pore for directional diffusion into the degradation chamber.

To identify the specific region of HspR that interacts with Bpa, we used a peptide-binding array (Fig 7). Our results showed that a conserved region of 15 amino acids located in the N-terminal region of the HTH domain was able to bind Bpa independently, and mutations in positively charged residues within this region led to a pronounced degradation defect (Fig 7E). Based on these results, we

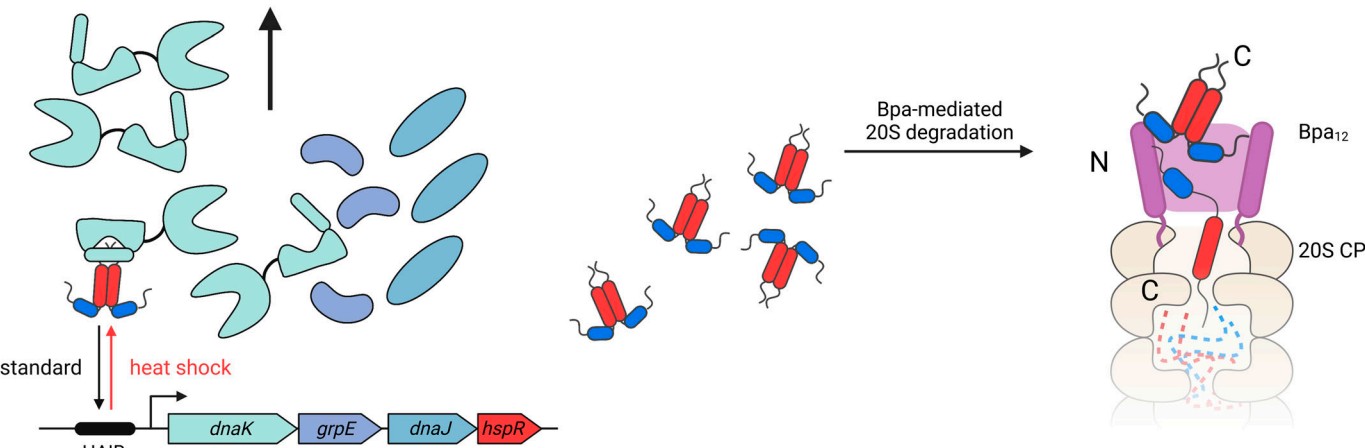

**Figure 8. Role of Bpa-dependent proteasomal degradation in *M. smegmatis* heat shock response.**
The proposed model shows the role of Bpa-dependent proteasomal degradation of HspR. Under standard conditions, the heat shock repressor HspR, in complex with its co-repressor DnaK, represses the expression of the *dnaKJgrpE-hspR* operon. Upon heat shock, HspR dissociates from its cognate DNA, the HAIR motif, and from DnaK. Free HspR is recruited for Bpa-meditated proteasomal degradation. Residues in the N-terminal HTH domain of HspR bind Bpa, whereas the C-terminal disordered tail of HspR acts as a threading motif to promote efficient proteasomal degradation.

propose a model for the ATP-independent proteasomal degradation of HspR during stress, where Bpa binds to the N-terminal region of the HTH domain, followed by threading of the disordered C-terminal region into the proteasome α-ring pore (Fig 8). Processive translocation of HspR into the degradation chamber is not driven by ATPase activity but likely relies on the dynamic folding/ unfolding equilibrium of HspR. With each unfolding event, HspR progresses deeper into the degradation chamber, while binding of the translocating polypeptide to the peptide-binding grooves surrounding the protease active sites prevents back-slippage.

In eukaryotes, ATP-independent proteasomal activators like PA200 have been proposed to facilitate the hydrolysis of small peptides and loosely folded proteins (Savulescu & Glickman, 2011). The exact role of PA200 is, however, still a subject of debate. Multiple hypotheses exist, suggesting it competes with ATP-dependent activators to regulate ubiquitin-dependent degradation, acts as a recruitment aid for the 26S-holo proteasome complex, or acts as a proteasomal activator itself, stimulating the degradation of small peptides or loosely folded proteins (Ustrell et al, 2002; Blickwedehl et al, 2007; Savulescu & Glickman, 2011; Chen et al, 2021). Although Bpa in bacteria shows no sequence or structural homology to ATP-independent proteasomal activators in eukaryotes, their functions may be analogous at least in some aspects.

The identification of natural substrates for ATP-independent proteasomal degradation has remained challenging, both in eukaryotes and bacteria. In eukaryotes, only a few substrates have been identified, such as peptides and non-native τ protein for PA200 (Schmidt et al, 2005; Dange et al, 2011; Huang et al, 2016). Similarly, in bacteria, only one other natural substrate (ParP) has been described in *M. tuberculosis*, but it can only be purified in a stable complex with Bpa, limiting its use for mechanistic in vitro studies (Hu et al, 2018).

Our study also provides insights into the role of Bpa in mycobacterial quality control in vivo. We found that Bpa is important for *M. smegmatis* during heat shock, consistent with previous observations in *M. tuberculosis* (Jastrab et al, 2015, 2017). A *bpa* deletion strain of *M. smegmatis* showed impaired adaptation to heat shock in liquid culture compared with the wild-type strain (Fig 1B). We propose a model where, under standard conditions, HspR is bound to its operator, the HAIR motif, which likely masks the Bpa-binding sites on the HspR HTH domain. During heat shock, HspR dissociates from DNA and its co-repressor DnaK, allowing for the efficient expression of the Hsp70 system to mitigate the effects of heat shock on the proteome (Fig 8). The newly expressed DnaK engages with proteins that display disordered, extended regions to prevent aggregation and misfolding. Meanwhile, the co-expressed HspR is degraded by the Bpa–CP complex to prevent accumulation and increased re-annealing of HspR to the HAIR motif, which would otherwise suppress the expression of vital heat shock chaperones and negatively impact viability. During heat shock recovery, when DnaK is no longer required for protein folding, free DnaK binds HspR, causing it to re-anneal to the HAIR motif, and this, in turn, represses the *dnaKJgrpE-hspR* operon.

Exploring a potential role of Bpa under other stress conditions, we discovered that Bpa also supports *M. smegmatis* adaption to oxidative stress, as the *bpa* deletion strain lost viability upon addition of $H_2O_2$ and complementation with *bpa* could prevent this phenotype (Fig 1C). It is plausible that Bpa and ATP-independent proteasomal degradation have broader roles in mycobacterial stress responses. As an ATP-independent proteasomal regulator, Bpa may be especially important in environments with limited ATP availability. Further in vivo studies are needed to fully understand the involvement of Bpa in mycobacterial stress responses and quality control beyond the degradation of HspR.

Our study also suggests a connection between the Bpa-mediated proteasomal degradation of HspR and the pupylation-dependent proteasomal degradation of HspR's co-repressor, DnaK. Previous studies on pupylated proteomes in actinobacteria identified DnaK as a potential pupylation target and substrate for ATP-dependent proteasomal degradation (Festa et al, 2010; Poulsen et al, 2010). In our study, we validated these findings in vitro, demonstrating that DnaK can be pupylated and subsequently degraded by the Mpa–proteasome system. Given that HspR represses the *dnaKJgrpE-hspR* operon, and DnaK acts as a co-repressor with HspR, the observed increased abundance of HspR in a *bpa* deletion strain of *M. smegmatis*, particularly under heat stress, raises interesting questions regarding a possible functional interplay between pupylation and proteasomal degradation of DnaK, Bpa-mediated degradation of HspR and the HspR-mediated suppression of the *hsp70* heat shock locus.

Taken together, our study emphasizes the importance of ATP-independent proteasomal degradation in mycobacteria for adaptation to environmental stress and provides insights into the substrate determinants for ATP-independent proteasomal degradation by the Bpa–CP complex. Further investigations are needed to determine if ATP-independent proteasomal degradation in mycobacteria is limited to a few specific substrates, such as HspR and ParP, or if the substrate selection is broader under specific stress conditions or involves other regulatory proteins that become substrates in specific environments.

# Materials and Methods

### Generation of the *bpa* deletion strain in *M. smegmatis*

Deletion of *bpa* in *M. tuberculosis* (Jastrab et al, 2015) was previously shown to affect the expression of essential downstream genes. Because *bpa* (MSMEG_6365) in *M. smegmatis* is located upstream of the orthologous genes (MSMEG_6366, MSMEG_6367, MSMEG_6369) and they were shown to be essential in *M. smegmatis* (Dragset et al, 2019), we followed an analogous strategy for knockout construction, first relocating the downstream genes before generation of the knockout (Parish & Brown, 2008). We constructed an MC²155 SMR5 strain carrying MSMEG_6366/6367/6369 on kanamycin resistant integrative plasmid (pTTP1B, Addgene #91723) under the natural promoter. This strain was then used as the parental strain to further knockout *bpa* along with MSMEG_6366/6367/6369 from their original genomic position. Up- and down-stream regions (1,500 base pairs each) of MSMEG_6365–6369, amplified from MC²155 SMR5 genomic DNA, was cloned into a hygromycin-resistant suicide plasmid (pGOAL19, Addgene #20188). 1 µg of suicide plasmid was treated with UV light for 1 min before transformation into strain MC²155 SMR5:MSMEG_6366/6367/6369 and plating onto 7H10 agar supplemented with 50 µg/ml hygromycin at 37°C. Single crossovers (SCOs), identified as blue colonies by X-gal (5-bromo-4-chloro-3-indolyl-*β*-D-galactopyranoside) underlay and overnight incubation at 37°C, were cultured in 7H9 at 37°C supplemented with 50 µg/ml hygromycin and grown to an OD600 of 0.7. Serial dilutions of $10^0$, $10^{-1}$, and $10^{-2}$ were plated onto 7H10 supplemented with 2% (wt/vol)

sucrose and incubated at 37°C until colonies grew. Double Cross-overs (DCOs) were identified as white colonies after X-gal underlay and overnight incubation at 37°C and were screened for *bpa* deletion by colony PCR. DCOs were restreaked onto 7H10 agar and single colonies of DCOs were further tested for absence of *bpa* by amplification of the genomic locus. DCOs that did not contain *bpa* (now confirmed *bpa* deletion strains) were grown in liquid 7H9 media to make glycerol stocks. The absence of *bpa* was also verified by Western blotting. One single DCO was used for this study and is referred to as the *M. smegmatis bpa* deletion strain. The *bpa* gene together with its natural promoter was amplified from MC²155 SMR5 genomic DNA and cloned into an apramycin resistant integrative vector (pFlag_attp, which was a gift from Markus Seeger, Addgene # 110095) to generate the complementation vector. The complemented strain was produced by co-transformation of this plasmid and a plasmid containing the integrase (pMA_int, which was a gift from Markus Seeger, Addgene # 110096) into competent *bpa* knockout *M. smegmatis* cells and plating onto 7H10 supplemented with 25 µg/ml kanamycin and 50 µg/ml apramycin. From a single verified colony, a glycerol stock of the *bpa*-complemented knockout strain was prepared.

### Bacterial strains and growth experiments

Bacterial strains used in this study were *M. smegmatis* MC²155 SMR5 (WT), *M. smegmatis* MC²155 SMR5 *bpa* deletion strain constructed as described above (Δ*bpa*) and the *bpa*-complemented *M. smegmatis* MC2155 SMR5 *bpa* deletion strain (Δ*bpa:bpa*). Bacterial cultures (WT, Δ*bpa*, and Δ*bpa:bpa*) were grown in M9 minimal medium, supplemented with 2 mM MgSO₄, 0.2% (vol/vol) glycerol, and 0.05% (wt/vol) Tween-80. For the heat stress growth experiment, strains were grown in triplicates in 30 ml cultures at 37°C to logarithmic phase and then diluted to an optical density at 600 nm of 0.01. Cells were further grown at 37°C under standard conditions or 45°C for heat shock. Optical density was measured at 600 nm at the given time points and plotted.

For oxidative stress experiments, strains were grown in triplicates in 30 ml cultures in the same minimal medium as mentioned above to logarithmic phase and then diluted to an optical density of 0.1. Cells were further grown under standard conditions or supplemented with 5 mM of $H_2O_2$ for oxidative stress. Optical density was measured at 600 nm at the given time points and plotted.

### Proteomics

25 ml bacterial culture were grown in 7H9 medium (Difco). Strains were grown at 37°C under standard conditions or stressed for 3 h at 45°C for heat stress. Cells were harvested and lysed by mechanical bead-beating in 20 mM HEPES–NaOH, pH 7.5, 150 mM KCl, 2 mM EDTA. Lysates were cleared by spinning for 5 min at 16,000*g*. Protein concentration of the lysates was determined by bicinchoninic acid assay (BCA Protein Assay Kit; Thermo Fisher Scientific) and diluted to 1 mg/ml. 50 µl of diluted lysate was mixed with sodium deoxycholate (DOC) to a final concentration of 5% (wt/vol). Proteins were reduced by incubation with 1,4-dithiothreitol (final concentration 12 mM) for 30 min at 37°C and alkylated by incubation with iodoacetamide (final concentration of 40 mM) for 45 min at room

temperature in the dark. Samples were diluted with 0.1 M ammonium bicarbonate to a final concentration of 1% (wt/vol) DOC. Proteins were digested overnight with lysyl endopeptidase (Wako Chemicals) and sequencing grade porcine trypsin (Promega) at an enzyme:substrate ratio 1:100 at 37°C with constant shaking (800 rpm). The digestion was stopped by the addition of formic acid to a final concentration of 1% (vol/vol) (pH <3). Precipitated DOC was filtered out by centrifugation at 800$g$ (Eppendorf rotor S-4x1000) for 5 min with a 0.2 $\mu$m PVDF membrane filter (Corning FiltrEX 96-well white filter plate). The peptide mixtures were loaded onto 96-well elution plates (Waters), desalted, and eluted with 80% (vol/vol) acetonitrile, 0.1% (vol/vol) formic acid. After elution, peptides were dried in a vacuum centrifuge, resolubilized in 0.1% (vol/vol) formic acid to a final concentration of 1 mg/ml and analyzed by mass spectrometry.

Data were acquired on an Orbitrap Eclipse Tribrid mass spectrometer in data-independent mode (DIA). The samples were separated using easy LC system by a 120 min linear gradient at a flow rate of 300 nl/min with increasing buffer B (95% (vol/vol) acetonitrile in 0.1% (vol/vol) formic acid) from 3% to 30% (vol/vol) on a 40 cm × 0.75 mm i.d. column (New Objective, PF360-75-10-N-5) packed in house with 1.9 $\mu$m C18 beads (Dr. Maisch Reprosil-Pur 120). The column was heated to 50°C. For DIA, a full-MS1 scan was acquired between 350 and 1,100 m/z at a resolution of 120,000 with an AGC target of 100%. Forty-one variable-width windows were used to measure fragmented precursor ions. DIA-MS2 spectra were acquired at a resolution of 30,000 and an AGC target of 400%. The first mass was fixed at 200 m/z and the normalized collision energy was set to 28. To maximize parallelization, a duty cycle time was 3 s.

The data were searched in Spectronaut version 14.11 (Biognosys), using a directDIA approach. The spectral library was created based on a pulsar search using the default setting and trypsin digestion rule. The data were searched against the UniProt FASTA database (*M. smegmatis*, strain ATCC 70084/mc(s)115, UP000000757, November 2018). The targeted data extraction was performed in Spectronaut version 14.11 with default settings except for the machine learning which was set to "across experiment" and the data filtering which was set to "Qvalue" (adjusted *P*-value). The FDR was set to 1% on peptide and protein level. Differential analysis on protein level was performed with Protti (Quast et al, 2022) package, using moderated *t* test and correcting for multiple hypothesis testing using Benjamini–Hochberg correction (Benjamini & Hochberg, 1995). Post processing and data visualisation were performed in R.

## Multiple sequence alignment and structural modelling

HspR amino acid sequences from several actinobacterial species were compiled and aligned in ClustalW (Larkin et al, 2007; Madeira et al, 2019). The multiple sequence alignment was visualized in Jalview (Waterhouse et al, 2009) and conservation was calculated by amino acid identity between residues. A structural model of HspR was predicted in AlphaFold (Jumper et al, 2021; Varadi et al, 2022) using the full-length amino acid sequence from *M. tuberculosis* HspR. This structural model was used in corroboration with information gained from the PASTA 2.0 prediction software (Walsh

et al, 2014) on regions of disorder, to construct a domain organization model of *M. tuberculosis* HspR.

## Cloning, expression, protein purification, and DNA probe generation

All genes for constructs for heterologous expression in *E. coli* were amplified by PCR from genomic DNA from either *M. tuberculosis* H37Rv or *M. smegmatis* MC²155 SMR5 with Q5 polymerase (New England Biolabs). For protein expression, coding sequences were cloned into a pET28a vector under a T7 promoter by Gibson assembly. Proteasome constructs were cloned into a pETDuet vector under a T7 promoter to allow for co-expression of the $\alpha$ and $\beta$ subunits, with a Strep-Tactin tag on the C-terminus of the $\beta$ subunit. All truncations and fusions were either generated by Gibson Assembly (New England Biolabs) or blunt end-ligation using T4 ligase (New England Biolabs). HspRΔN15, HspR E19A, HspR E19K, and HspR C-terminal fusions were fused to a histidine tag containing maltose-binding protein (His$_6$MBP) at the N-terminus. All constructs were transformed into *E. coli* Rosetta cells for overexpression and induced using an autoinduction media system (Studier 2005). Cells were lysed with a high-pressure homogenizer (Microfluidizer M110-L, Microfluidics).

His$_6$MBP-HspR variants, His$_6$MBP-HspR fusions, His$_6$Bpa, and His$_6$DnaK were purified by Ni-NTA affinity chromatography. Proteasome variants were isolated by StrepTactinXT affinity chromatography. Bpa and proteasome variants were applied directly onto a Superdex 200 or Sepharose 6 size exclusion chromatography column, respectively, and purified in 50 mM HEPES–NaOH, pH 7.5, 150 mM NaCl, 1 mM EDTA, 10% (vol/vol) glycerol. The other proteins were first cleaved with Tobacco Etch Virus endopeptidase (TEV protease) and further purified by reapplication to Ni-NTA affinity chromatography to remove the His$_6$ tag and/or application to amylose chromatography to remove the His$_6$–MBP fusion from the protein of interest. HspR variants were applied onto a heparin column and run in a linear gradient from 50 mM HEPES–NaOH, pH 7.5, 0 M–2 M NaCl, 5 mM MgCl$_2$, 5% (vol/vol) glycerol to isolate the protein of interest. Finally, proteins were then applied onto a Superdex 75 size exclusion chromatography column and purified in 50 mM HEPES–NaOH, pH 7.5, 150 mM NaCl, 5 mM MgCl$_2$, 5% (vol/vol) glycerol.

FimAt fusion-variants, HspR WT, HspRΔC9, and HspR H24A were isolated from inclusion bodies under denaturing conditions. Cells were resuspended and lysed as mentioned above and then added to 0.8 x volume of Triton buffer (60 mM EDTA 1.5 M NaCl, adjusted to pH 7.0, 6% [vol/vol] Triton X-100), stirring at 4°C for 30 min. Inclusion bodies were harvested by centrifugation (20,000 rpm for 45 min) and washed five times in 100 mM Tris–HCl, pH 8.0, 20 mM EDTA. Inclusion bodies were solubilized in 5 ml of buffer (50 mM Tris, 6 M guanidinium chloride) per gram of inclusion bodies by stirring for 2 h at room temperature. Solubilized inclusion bodies were spun at 100$g$ for 30 min at 20°C. The supernatant containing solubilized protein was refolded by dialysis against 50 mM Tris–HCl, pH 7.5, 5 mM MgCl$_2$, 0.5 M arginine, 10% (vol/vol) glycerol. Refolded proteins were further purified by ion exchange chromatography (50 mM HEPES–NaOH, pH 7.5, 50 mM to 1 M NaCl, 5 mM MgCl$_2$, 5% [vol/vol] glycerol) and gel filtration over a Superdex 75 size exclusion column

in 50 mM HEPES–NaOH, pH 7.5, 150 mM NaCl, 5 mM MgCl$_2$, 5% (vol/vol) glycerol. Mpa, PafA, PanB, and Pup were purified as described previously (Laederach et al, 2019) and purified FimAa and FimAt were kindly gifted by the Glockshuber laboratory. The HAIR motif was generated as previously described (Parijat & Batra, 2015). DNA was produced was by annealing 100 $\mu$M of forward (fw) and reverse (rv) primers at 95°C in 70 cycles, decreasing 1°C/cycle until 25°C. The primers were the following: HAIR motif oligomers (fw 5′-CCG TCG AGG CAA GCT TGA GCG GGG TGC ACT CAT CAT AGT GCA GGA AAG AA-3′); (rv 5′-TTC TTT CCT GCA CTA TGA TGA GTG CAC CCC GCT CAA GCT TGC CTC GAC GG-3′).

## RNaseA alkylation

To permanently unfold model substrate ribonuclease A (RNaseA), lyophilized protein from bovine pancreas was purchased (Sigma-Aldrich) and dissolved at a concentration of 3 mg/ml in degassed denaturing buffer (6 M guanidinium chloride, 10 mM Na$_2$EDTA, 200 mM Tris–HCl, pH 8.6, 1 mM DTT) at room temperature for 30 min. In the dark, 300 $\mu$l reaction, containing 200 $\mu$l of the 3 mg/ml denatured and reduced RNaseA was incubated with 17 mM iodoacetamide (Sigma-Aldrich) in denaturing buffer at room temperature for 30 min. The denaturing buffer was exchanged by dialysis or by application over a PD10-desalting column (GE Healthcare) with 20 mM sodium phosphate buffer (pH 8.0). As cysteine alkylation with iodoacetamide results in addition of a carbamidomethyl group (57 D), full alkylation of all eight cysteine residues was confirmed by intact mass spectrometry.

## Circular dichroism and thermal transitions

To analyze protein fold and stability, we used circular dichroism (CD) to measure secondary structure content. Model substrates were dialyzed into 20 mM sodium phosphate buffer (pH 8.0) before CD measurements. For thermal transitions, HspR variants were dialyzed into 20 mM sodium/potassium buffer, pH 7.5, 150 mM NaCl, 1 mM MgCl$_2$. CD spectra were recorded by measuring ellipticity ($\Theta$) in a 0.1 cm quartz cuvette (Hellma Analytics) placed in a Jasco J-710 spectropolarimeter (Brechbühler). For CD spectra, the measured ellipticity ($\Theta$) from five acquisitions was normalized to mean molar ellipticity in degree cm$^2$ dmol$^{-1}$:

$$MRW = \frac{\Theta * 100 * kD}{c * d * N}$$

where N is the number of amino acid residues, d is the cuvette pathlength in cm, and c is the protein concentration in mg/ml.

Thermal transitions by CD spectroscopy for HspR and variants were measured at 222 nm from 20°C to 90°C. The recorded ellipticity was first fitted with the following equation for a one-state thermal transition:

$$S_{obs} = \frac{\left[ (S_n^\circ + m_n \bullet T) + (S_u^\circ + m_u \bullet T) \exp\left(\frac{\Delta H_m}{R \bullet T_m}\right)\left(\frac{T - T_m}{T_m}\right) \right]}{\left[ 1 + \exp\left(\frac{\Delta H_m}{R \bullet T_m}\right)\left(\frac{T - T_m}{T_m}\right) \right]}$$

where S$_{obs}$ is the observed ellipticity, $S_u^\circ$ is ellipticity of unfolded protein extrapolated to 0 K, m$_u$ is the slope of the unfolded protein

ellipticity, $S_n^\circ$ is the ellipticity of native protein extrapolated to 0 K, m$_n$ is the slope of the native protein ellipticity, and T is the temperature in Kelvin, T$_m$ is the melting temperature, $\Delta H_m$ is the enthalpy difference between native and unfolded protein at T$_m$, R is the gas constant, and T is the temperature in K. For thermal transitions, experiments were performed in independent triplicates and one representative transition is shown.

Next, ellipticity was converted to the fraction of folded HspR (f$_N$) according to the following equation taking into account the linear temperature dependencies of pre- and post-transition baselines, using the parameters determined by fitting with the equation mentioned above:

$$f_N = \frac{S_{obs} - (S_u^\circ + m_u \bullet T)}{(S_n^\circ + m_n \bullet T) - (S_u^\circ + m_u \bullet T)}$$

where S$_{obs}$ is the observed ellipticity, $S_u^\circ$ is ellipticity of unfolded protein extrapolated to 0 K, m$_u$ is the slope of the unfolded protein ellipticity, $S_n^\circ$ is the ellipticity of native protein extrapolated to 0 K, m$_n$ is the slope of the native protein ellipticity, and T is the temperature in Kelvin.

## Proteasomal degradation assays

4 $\mu$M substrate was incubated with 12 $\mu$M Bpa protomer or 200 $\mu$M hexapeptide (GTGQYL) and 0.4 $\mu$M assembled proteasome core particle (20S CP$^{WT}$ or 20S CP$^{OG}$) in 50 mM HEPES–NaOH, pH 7.5, 150 mM NaCl, 5 mM MgCl$_2$, 1 mM DTT, and time points were taken at the indicated intervals. As a negative control, Bpa was omitted from the reaction and as a positive control substrate was incubated with 20S CP$^{OG}$ in absence of Bpa. To quantify degradation, band intensity was measured from triplicate gels via densitometry (GelAnalyzer 19.1) and plotted against time. Reactions were incubated at 37°C and time points were taken at the indicated intervals. To investigate the temperature dependence of certain degradation defects, reactions were, in addition to 37°C, incubated at 30°C and 42°C and indicated as such in figures.

To test if HspR is a Bpa-specific substrate, 3 $\mu$M Mpa$_6$, or Bpa$_{12}$ were incubated with 0.4 $\mu$M 20S CP$^{OG}$ in the presence of 2.5 mM ATP for 30 min at room temperature to assemble the Bpa– or Mpa– proteasome complexes. After the pre-incubation, 2.5 mM additional ATP was added, and the reactions were started by adding 4 $\mu$M substrate. Reactions were conducted in 50 mM HEPES–NaOH, pH 7.5, 150 mM NaCl, 5 mM MgCl$_2$, 1 mM DTT. Reactions were incubated at 37°C and time points were taken at the indicated intervals.

To measure the initial velocity of HspR degradation, we conducted the Bpa-mediated proteasomal degradation assay at closer time intervals in the early phase of degradation (0–12 min for HspR WT and 0–2 h for HspR$\Delta$C). Speed of degradation was assessed densitometrically by measuring band intensity decrease of HspR over time (GelAnalyzer 19.1). Reactions were incubated at 37°C, and time points were taken at the indicated intervals.

To determine if pupylated DnaK from *M. tuberculosis* is a proteasomal degradation substrate, 2 $\mu$M DnaK–Pup was incubated with 0.2 $\mu$M 20S CP$^{OG}$, 0.2 $\mu$M Mpa$_6$ in 50 mM HEPES–KOH pH 7.5, 150 mM NaCl, 10 mM KCl, 20 mM MgCl$_2$, 10% (vol/vol) glycerol, 10 mM

ATP, 1 mM DTT. Reactions were incubated at 37°C and time points were taken at the indicated intervals.

For all degradation assays, time points were quenched in Laemmli buffer and reactions were visualized by running samples on SDS–PAGE. For all gel-based degradation assays, reactions were conducted in independent triplicates and one representative gel is shown.

### Pupylation assays

6 $\mu$M of substrate was incubated with 12 $\mu$M Pup and 1 $\mu$M PafA in buffer P (50 mM HEPES–KOH, pH 8.0, 150 $\mu$m NaCl, 20 mM MgCl$_2$, 10% [vol/vol] glycerol, 1 mM DTT, 5 mM ATP). Reactions were incubated at 30 °C and time points were taken at the indicated intervals. Time points were quenched in Laemmli buffer and reactions were visualized by running sub samples on SDS–PAGE. For all gel-based pupylation assays, reactions were conducted in independent triplicates and one representative gel is shown.

Pupylated DnaK from *M. tuberculosis* was produced in batch by incubating 35 $\mu$M DnaK with 40 $\mu$M Pup, and 1 $\mu$M PafA in buffer P for 20 h at 30 °C. Pupylated DnaK was separated from other proteins via gel filtration on a Superdex 200 column.

### Substrate-binding assays

Affinity of substrate for Bpa was investigated by measuring the temperature-related fluorescence intensity changes in a Monolith NT.115 from NanoTemper. First, Bpa lysine residues were labeled with a 20-fold excess of fluorescein. Bpa and dye were incubated for 1 h at room temperature, after which the mixture was spun at 14,000 rpm for 10 min at 4°C. Excess fluorescein was separated from labeled Bpa (Bpa-Fl) by gel filtration over a Superdex 200 analytical size exclusion column in 50 mM HEPES–NaOH, pH 7.5, 150 mM NaCl, 10% (vol/vol) glycerol.

For binding experiments, 200 nM Bpa-Fl protomer was titrated with serial dilutions of HspRΔC9, HspR WT, or HspR H24A at indicated concentrations in 50 mM HEPES–NaOH, pH 7.5, 150 mM NaCl, 5 mM MgCl$_2$, 5% (vol/vol) glycerol, 0.01% (vol/vol) Tween-20. Binding of HspR WT to Bpa was either measured with HspR WT alone or in a 1:1 complex with DnaK or the HAIR operator DNA. Reactions were pre-incubated 10 min at room temperature and absorbed into premium capillary tubes (NanoTemper). The measurements on the Monolith NT.115 were performed at 20°C with 20% blue LED power and 20%, 40%, and 80% MST power, which corresponded to a temperature jump of 2–8°C. For each substrate, the binding titration was conducted at least in triplicate, and scans were compiled and analyzed with the NTAffinity Analysis v2.0.2 software using the T-jump model to calculate dissociation constants based on the Kd model (Jerabek-Willemsen et al, 2014). Error bars denote the SD.

### Peptide-binding array

To identify regions of *M. tuberculosis* HspR contributing to Bpa binding, we designed peptides of 15 amino acids in length that covered the sequence of HspR excluding the N- and C- terminal stretches (Table S3). In total, eight peptides were ordered (Genescript) and dissolved in buffer M (50 mM HEPES–KOH, pH 7.5, 150 mM

NaCl, 5 mM MgCl2, 5% [vol/vol] glycerol, 0.1% [wt/vol] pluronic 127, 2% [vol/vol] DMSO). Bpa lysine residues were labeled with threefold excess RED-NHS second generation dye using the RED-NHS second generation kit according to manufacturer guidelines (NanoTemper).

Binding experiments were conducted on a Monolith NT.115 instrument (Nanotemper). 250 nM Bpa-RED protomer was titrated with serial dilutions of 2 mM–30 nM peptide 1–7 in buffer M. Reactions were pre-incubated 10 min at room temperature and aspirated into premium capillary tubes (NanoTemper). The measurements were performed on a Monolith NT.115 (NanoTemper) at 25°C with 20% red LED power and 40% MST power. Scans were compiled and analyzed with the NTAffinity Analysis v2.0.2 software using the T-jump model to calculate dissociation constants based on the Kd model (Jerabek-Willemsen et al, 2014).

## Data Availability

Proteomic data have been uploaded to the ProteomeXchange Consortium via the PRIDE partner (Project accession: PXD039021).

## Supplementary Information

## Acknowledgements

We would like to thank the Functional Genomics Center Zurich and Jens Sobek for conducting the intact mass spectrometry experiment and providing facilities for the binding experiments, respectively. This work was supported by a grant of the Swiss National Science Foundation (31003A_163314) and an ETH research grant ETH-17 17-2.

### Author Contributions

T von Rosen: conceptualization, data curation, formal analysis, validation, investigation, visualization, methodology, and writing—original draft, review, and editing.
M Pepelnjak and J-P Quast: data curation, formal analysis, and writing—review and editing.
P Picotti: supervision, investigation, and writing—review and editing.
E Weber-Ban: conceptualization, supervision, funding acquisition, writing—original draft, review, and editing, and project administration.

### Conflict of Interest Statement

The authors declare that they have no conflict of interest.

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
