## [Reviewer comments · Life Science Alliance]

Life Science Alliance

ATP-independent substrate recruitment to proteasomal degradation in mycobacteria

Tatjana von Rosen, Monika Pepelnjak, Jan-Philipp Quast, Paola Picotti, and Eilika Weber-Ban

DOI: <https://doi.org/10.26508/lsa.202301923>

Corresponding author(s): Eilika Weber-Ban, ETH Zurich

Review Timeline:

Submission Date:	2023-01-13
Editorial Decision:	2023-02-27
Revision Received:	2023-06-30
Editorial Decision:	2023-07-24
Revision Received:	2023-07-28
Accepted:	2023-07-31

Scientific Editor: Novella Guidi

Transaction Report:

February 27, 2023

Re: Life Science Alliance manuscript #LSA-2023-01923-T

Prof. Eilika Weber-Ban
ETH Zurich
Institute for Molecular Biology and Biophysics
Otto-Stern-Weg 5
Zurich 8093
Switzerland

Dear Dr. Weber-Ban,

Thank you for submitting your manuscript entitled "Pupylation-independent substrate recruitment to proteasomal degradation in mycobacteria" to Life Science Alliance. The manuscript was assessed by expert reviewers, whose comments are appended to this letter. We invite you to submit a revised manuscript addressing the Reviewer comments.

Thank you for this interesting contribution to Life Science Alliance. We are looking forward to receiving your revised manuscript.

Sincerely,

B. MANUSCRIPT ORGANIZATION AND FORMATTING:

Reviewer #1 (Comments to the Authors (Required)):

Mycobacteria, such as the human pathogen *Mycobacterium tuberculosis*, utilize chaperone-protease systems as well as an eukaryotic-like proteasome for selective protein degradation. Proteasomal function requires interaction with adaptors --oligomeric rings of the unrelated Mpa or Bpa proteins. Whereas Mpa has ATPase activity and binds to substrates tagged with the post-translational modifier Pup, how Bpa selects and processes substrates has remained unknown.

In this manuscript, Ban and colleagues investigate the function of Bpa. They begin by showing that Bpa is critical for the survival of *Mycobacterium smegmatis* under stress conditions that are associated with the accumulation of misfolded or damaged proteins, such as heat or oxidative stress. They next follow up with a series of experiments that uncover mechanisms underlying the cellular phenotype of bpa mutants. The authors show that HspR, a DNA-binding protein that acts as negative regulator of chaperone gene transcription, is a major substrate of the Bpa-mediated degradation in vivo. Elegant biochemical experiments reconstituting Bpa-mediated HspR degradation in vitro uncover specificity determinants. Overall, the data suggest that Bpa binds to HspR via the folded N-terminal HTH domain, while the HspR unstructured C-terminal tail facilitates proteasomal degradation. The authors also find that the HspR interactor, heat shock chaperone DnaK, is a substrate of the Mpa-proteasome complex.

This is a nice piece of work that opens a new avenue for understanding and further study of mycobacterial protein degradation and stress response. I strongly recommend publication in Life Science Alliance. No further changes are required for publication. For future studies, I suggest a few directions the authors could try to boost evidence for the proposed mechanism:

1. The results imply that HspR is the critical Bpa substrate in the response to stress. Can the authors engineer HspR mutations that prevent Bpa interaction yet enable the protein to remain functional as a transcriptional repressor, and show that the mutant HspR renders cells sensitive to stress in the presence of functional Bpa?
2. The authors hypothesize that binding to its operator DNA stabilizes HspR against proteasomal degradation, and nicely show that degradation of HspR in vitro was inhibited by the presence of DNA containing the HspR binding motif (HAIR). Future work should develop the assay further. For example, is degradation of a DNA binding-defective HspR mutant insensitive to the presence of the DNA oligo? Conversely, is a mutant DNA oligo that fails to bind HspR likewise inactive in stabilizing HspR in vitro? Does the HAIR oligo preparation inhibit 20S activity, degradation of an Mpa substrate, or degradation of HspRΔC9 by the 20S CPOG? Is the oligo concentration required to observe a stabilizing effect consistent with the K_d for HspR binding? Such experiments would strengthen the conclusion that the observations are due to a specific effect of the DNA, as the authors have nicely shown for the HspR-DnaK interaction.
3. The finding that the unstructured C-terminus of HspR is required for degradation is exciting and adds to the growing literature related to proteolytic functions of C-terminal sequences. I would be curious to know whether sequences such as the Alanine tails made during bacterial ribosome-associated quality control (RQC) would suffice to restore degradation activity to a C-terminal-deleted HspR.

I stress that these comments are meant to be suggestions to add further value to the review process, and not a requirement for publication. I leave it to the authors' discretion whether to perform any additional experiments.

Reviewer #2 (Comments to the Authors (Required)):

In this manuscript, the authors aim to shed some light on the mechanisms of substrate recognition/recruitment by the Bpa-mediated proteasomal degradation system, one of the two alternative pathways of proteasomal degradation in mycobacteria and other Actinobacteria. While the degradation pathway involving the Mpa ATPase, an activator that recognizes Pup-modified substrates, has been fairly well studied, there is limited knowledge on the Bpa pathway, which requires neither energy nor pupylation for target substrate degradation.

By studying the transcriptional repressor HspR, a natural substrate for Bpa, as well as two conformational model substrates that are locked in either native or non-native states, the authors went on to conclude that conformational disorder is required for a substrate to enter the 20S proteasomal chamber for subsequent degradation, though not sufficient for recognition by Bpa. In the case of HspR, the latter is seemingly mediated by an N-terminal sequence stretch, as the authors show; and subsequent to this, the C-terminally disordered region likely mediates threading into the chamber.

Major comment:

Overall, this is an interesting study with sufficient novelty. My major concern is that the manuscript seems to consist of several independent parts (substories) that have failed to mature into complete stories on their own, but were rather put together in an incoherent manner that eventually disrupts the flow and prevents the authors from giving a clear and focused message. For example, the authors start with a proteomic analysis to identify proteins whose abundances change upon Bpa deletion, both under standard and heat shock conditions. They identify several proteins, yet they don't follow up any of them. This specific experiment seems to serve merely to confirm some of the previous findings in *M. tuberculosis* this time in *M. smegmatis*; specifically, corroborating a role for Bpa in HspR turnover. In fact, it would have been very nice to identify novel substrates for Bpa (which are potentially among their listed proteins) and to try and search for commonalities between them (i.e. conformationally disordered regions) that may facilitate recognition/degradation by the Bpa-20S system.

Similarly, the authors demonstrate a role for Bpa for bacterial survival during oxidative stress. This phenotype is very strong (even more so than heat shock), but the authors don't follow this up either. I am not sure how these findings with oxidative stress (or even heat shock) fit in the entire story of the manuscript. The authors could have performed proteomic studies also under oxidative stress conditions and try to identify protein(s) whose abundances change in Bpa-deleted bacteria.

Then the story takes a completely different turn; using HspR and two 2 model substrates, the authors try to identify substrate determinants for binding to and degradation by Bpa-20S.

Finally, in Figure 8, the authors proceed once again on a different axis, this time trying to prove that DnaK is a pupylation target and degraded by the Mpa-20S system. Although I understand the authors' intention here (which is to demonstrate a potential interplay between the Bpa-20S system and the Mpa-20S system in regulating the *dnaKJgrpE-hspR* operon), the experiments in Figure 8 are not sufficient to establish this idea of interplay at a physiologic level. Thus, I feel that the idea in Fig 8 is immature, underdeveloped and probably out of context here.

To sum up, I feel that a clear and focused message is lacking. The authors try to develop this message around the substrate determinants that mediate Bpa binding and 20S-induced degradation, but in this case both Figure 1 and Figure 8 remain as underdeveloped substories that are potentially irrelevant to this central message.

Specific comments:

In Figure 1, the authors corroborate some of the previous knowledge obtained in *M. tuberculosis* this time in *M. smegmatis*. Specifically, in *bpa* knockout bacteria, they perform proteomic analyses to identify the proteins whose abundances change (when compared with wild type bacteria) both under standard and stress conditions. This experiment shows that HspR accumulates 3-fold in the absence of Bpa even in standard conditions, and 5-fold when cells are exposed to heat shock, suggesting that HspR may indeed be a substrate for the Bpa-20S proteasomal degradation pathway. The authors also report on their discovery of several other proteins whose abundances change upon *bpa* deletion, but they never revisit or follow up any of them.

Because only a few substrates have been identified for ATP-independent proteasomal degradation across species, in my opinion, it would have been interesting to pick another candidate from this list to further study it (i.e. is this protein a novel substrate for Bpa-20S in vitro?).

The authors also show that under stress conditions (heat shock: expected; oxidative stress: maybe surprising and novel) the absence of *bpa* is disadvantageous for the bacteria. Because the phenotype with oxidative stress is very strong (total growth impairment in the absence of *bpa*), it would have been interesting to further explore this. For example, would it be possible to perform another proteomic analysis in *bpa* knockout bacteria exposed to oxidative stress to see which proteins accumulate (in comparison with wild type)? This would allow the authors to potentially identify novel Bpa substrates that allow bacteria to survive oxidative stress conditions. They could then check whether this novel substrate also has disordered regions that are essential for Bpa-20S-mediated degradation.

In Figure 2, the authors perform in vitro proteasomal degradation assays on HspR, either when it is naked, or after pre-incubation with its target DNA sequence or with its co-repressor DnaK. These assays reveal that HspR, when bound to either of these (likely preventing it from being partially unfolded), is resistant to Bpa-20S-mediated degradation. In addition, here the authors also show that Mpa does not target HspR to 20S, confirming the specificity of Bpa towards this substrate. Although these results are potentially convincing, quantifications of the band intensities (densitometric analyses) from at least 3 such experiments represented in graph would be helpful.

In Figure 3, the authorz aligned the amino acid sequences of 20 different HspR orthologs, and also determined the alpha fold structure of the HspR dimer from *M. tuberculosis*. Eventually, they identified the regions of conservation (in the DNA-binding HTH domain, as well as towards the C-terminus) and predicted that the amino and carboxy termini of the protein contains conformationally disordered regions.

To further study the potential roles of these regions of disorder, they went on to delete them to create the deltaN15 or deltaC9 mutants. The experiments in Figure 4 show that the deletion of the C-terminal disordered region indeed impairs Bpa-20S-mediated degradation. Here, the densitometric analyses are provided and represented on a nice graph.

In Figure 5, to investigate how "disordered conformations" actually contribute to degradation, the authors use 2 model substrates either in their native state (RNaseA or FimAa) or locked in their non-native conformations (unfolded alkylated RNaseA or FimAt). As expected, neither the native RNaseA nor the permanently folded FimAa are degraded by the Bpa-20S system, or even by the engineered open gated proteasome. However, surprisingly, unfolded RNaseA or permanently unfolded FimAt are not degraded by Bpa-20S, either. With this observation, the authors suggest that conformational disorder is not sufficient to make a protein a substrate for Bpa-20s. Interestingly, the latter two are actually degraded by the open gated proteasome, suggesting that disorder is in fact required for entry into the 20S chamber. Here my only problem is with Figure 5D. The authors claim that FimAt fused with the C-terminal stretch of HspR is not degraded efficiently, but this is difficult to see on the gel image. Can they provide quantifications here as well?

In Figure 6, the authors perform binding assays using fluorescently labeled Bpa, and aim to test whether the disordered C-terminal part of HspR is involved in Bpa binding. In this assay, they record the binding-initiated changes in the fluorescent intensity, using either WT HspR or the DeltaC9 mutant. Eventually, they conclude that both of these bind with the same affinity to Bpa. Although this is a beautiful biophysical assay, it would be nice if it was complemented with some robust biochemistry, such as co-immunoprecipitation. In Figure 6D, the authors perform rescue experiments to see whether they can restore the degradation of DeltaC9 HspR by fusing it with two different 9 aa long linkers, which do not seemingly help. Looking at their gels, this conclusion is hard to reach. Either better gel images or densitometric quantifications must be presented here, because the bottom two panels actually show that the two linkers indeed promote degradation (maybe not very efficiently, but still to a significant extent). The authors should also clarify what a "GS linker" is.

In Figure 7, by performing a similar binding assay using fluorescently labeled Bpa and 8 short peptides covering most of the HspR sequence, the authors discover that the region corresponding to amino acids 16-30 mediates HspR binding to Bpa. By performing site directed mutagenesis, they showed that Glu19 and His24 were important for Bpa-20S-mediated degradation.

First off, the authors should clarify why they specifically focused on these two amino acids.

Secondly, they should investigate the binding efficiency (or lack thereof) of H24A, E19K and E19A mutants (either in the context of peptide 1 or in the context of the entire HspR protein) to fluorescently labeled Bpa. This is important and would complement their degradation experiment in Fig 7E. In principle, binding should also be impaired.

Looking at Fig 7A, degradation of the H24A, E19K and E19A mutants is not fully impaired. How do the authors explain this? Do other amino acids in this stretch compensates for the ablation of H24 or E19? Also, what if the authors combined their deltaC9 mutation with H24A, E19K or E19A? Would that fully abolish degradation?

Finally, the authors should test whether the H24A, E19K and E19A mutants are degraded by the open gated proteasome. In principle, if the point mutations impair binding with Bpa, even the open gated proteasome would fail to degrade the mutants. I think it would be nice to show this.

In Figure 8, the authors mention in the text that they have identified (by mass spectrometry) 4 residues on DnaK that are modified by Pup. However, I cannot see any data that support this statement, neither in the main figures nor in supplementary data.

Minor comments:

Overall, this is an interesting study. As a minor comment, I think that the discussion part is too long and contains a lot of repetitions of the results part. This should be avoided. Also, the authors claim in several places that the C-terminal disordered stretch of HspR is involved in threading towards the 20S core. Although some of the data may suggest this, definitive proof is still lacking, so the authors should try to tone down a little bit.

Reviewer #3 (Comments to the Authors (Required)):

Protein degradation by the proteasomal 20S CP particle in bacteria is assisted by ATP-dependent Mpa or Bpa, which functions independent of nucleotides. Bpa was shown before to play a role in stress resistance and to degrade the heat shock regulator HspR, which controls expression of the bacterial Hsp70 (DnaK) chaperone machinery. Here the authors mechanistically dissected how Bpa targets HspR for degradation by 20S CP. They identify two distinct N- and C-terminal regions in HspR that independently provide sequence (N-term) and conformational (C-term) selectivity. Thus Bpa functions not solely as activator of 20S CP but it additionally selects specific substrates for ATP-independent proteolysis. Furthermore, protein disorder is necessary but not sufficient for Bpa-mediated degradation, underlining that Bpa functions as substrate selector. This mechanistic dissection is very well documented and provides important insights into protein degradation by Bpa-20S CP complexes. The manuscript is therefore highly suitable for publication in Life Science Alliance. There are, however, a few points that the authors should consider in a revised manuscript.

Main points:

Deletion of Bpa causes stabilization of the heat shock repressor HspR (Fig. 1). The stress-sensitive phenotype of *bpa* cells can therefore be explained by an insufficient heat shock response, which does not allow synthesis of Hsp70 at sufficient levels (Fig. 9). While this model is appealing, no experimental evidence is provided here. Do the authors observe reduced accumulation of Hsp70 in *bpa* cells after heat shock in comparison to wild type cells?

Fig. 2A: The presence of the HAIR DNA operator protects HspR from degradation by Bpa/20S CP. The interaction between HspR and HAIR is believed to be temperature-sensitive, leading to HspR dissociation upon heat shock. Can the authors reconstitute such temperature-dependency in vitro? Can HspR be degraded despite HAIR presence at increased temperatures?

Fig.3: The sequence determinants of the C-terminal tail, which are crucial for its impact on HspR stability and degradation, remain undefined. According to the alignment hydrophobic and basic residues seem conserved. Can the authors speculate on the relevance of hydrophobicity and positive charge for C-tail function?

Fig.6C: The C-terminal tail reduces the thermal stability of HspR. Can the degradation defects of HspR- C be compensated by performing the experiments at increased temperatures (e.g. 48{degree sign}C) which cause HspR- C destabilization?

Fig. 8: The relevance and specificity of Hsp70 (DnaK) pupylation in vitro is questionable. Pupylation of DnaK is extremely slow in comparison to the model substrate PanB and thus not convincing. This result is also not connected to the remaining data focusing on Pup-independent HspR degradation. It is therefore recommended to remove Fig. 8 and the respective section from the manuscript.

The discussion section is very long and sometimes just repeating findings of the results part. A sharpening and shortening of the discussion is therefore recommended.

We would like to thank the reviewers for their time, their careful reading of our manuscript and their insightful comments. Point-by-point answers to the comments are found below.

Response to Reviewer 1

This is a nice piece of work that opens a new avenue for understanding and further study of mycobacterial protein degradation and stress response. I strongly recommend publication in Life Science Alliance. No further changes are required for publication. For future studies, I suggest a few directions the authors could try to boost evidence for the proposed mechanism:

1. The results imply that HspR is the critical Bpa substrate in the response to stress. Can the authors engineer HspR mutations that prevent Bpa interaction yet enable the protein to remain functional as a transcriptional repressor, and show that the mutant HspR renders cells sensitive to stress in the presence of functional Bpa?

This is an interesting idea and would provide nice *in vivo* support for our findings. However, it is difficult to design HspR mutations that impact only the binding to Bpa and/or its processing by Bpa, because transcriptional regulation involves both binding of DNA and of Hsp70. We will keep this in mind for future experiments but believe this to be out of the scope for work presented in this manuscript.

2. The authors hypothesize that binding to its operator DNA stabilizes HspR against proteasomal degradation, and nicely show that degradation of HspR *in vitro* was inhibited by the presence of DNA containing the HspR binding motif (HAIR). Future work should develop the assay further. For example, is degradation of a DNA binding-defective HspR mutant insensitive to the presence of the DNA oligo? Conversely, is a mutant DNA oligo that fails to bind HspR likewise inactive in stabilizing HspR *in vitro*? Does the HAIR oligo preparation inhibit 20S activity, degradation of an Mpa substrate, or degradation of HspR Δ C9 by the 20S CPOG? Is the oligo concentration required to observe a stabilizing effect consistent with the K_d for HspR binding? Such experiments would strengthen the conclusion that the observations are due to a specific effect of the DNA, as the authors have nicely shown for the HspR-DnaK interaction.

We have constructed a DNA-binding-defective mutant, HspR R29A and show that it is degraded by the Bpa-CP complex, both in the presence and absence of the target DNA. The rate of degradation is still slightly slowed in presence of the DNA, however this is likely due to unspecific binding of the DNA. We have added this data to the supplemental figure section (Fig S5A and B). We also found that HspR Δ C9, in complex with the target DNA, could not be degraded by the 20S CP^{OG} (Fig S5C). This suggests that when HspR is in a stable complex with the HAIR motif it cannot be degraded by the proteasome. However, the degradation defect is not due the presence of DNA inhibiting the proteasome, as the HspR R29A mutant is still readily degraded by Bpa-CP in the presence of target DNA. This data is also included in Fig S5.

Measurements of temperature-related fluorescence intensity changes of Bpa titrated with HspR alone or HspR in presence of target DNA shows that Bpa binds HspR less tight in presence of DNA, although slightly better than HspR-DnaK S1A. We have added the HspR-DNA binding to Bpa data to the supplemental Figure S1A .

3. The finding that the unstructured C-terminus of HspR is required for degradation is exciting and adds to the growing literature related to proteolytic functions of C-terminal sequences. I would be curious to know whether sequences such as the Alanine tails made during bacterial ribosome-associated quality control (RQC) would suffice to restore degradation activity to a C-terminal-deleted HspR.

This is an interesting question. We will for sure take this into consideration for future experiments.

I stress that these comments are meant to be suggestions to add further value to the review process, and not a requirement for publication. I leave it to the authors' discretion whether to perform any additional experiments.

Response to Reviewer 2:

Major comment:

Overall, this is an interesting study with sufficient novelty. My major concern is that the manuscript seems to consist of several independent parts (substories) that have failed to mature into complete stories on their own, but were rather put together in an incoherent manner that eventually disrupts the flow and prevents the authors from giving a clear and focused message. For example, the authors start with a proteomic analysis to identify proteins whose abundances change upon Bpa deletion, both under standard and heat shock conditions. They identify several proteins, yet they don't follow up any of them. This specific experiment seems to serve merely to confirm some of the previous findings in M. tuberculosis this time in M. smegmatis; specifically, corroborating a role for Bpa in HspR turnover. In fact, it would have been very nice to identify novel substrates for Bpa (which are potentially among their listed proteins) and to try and search for commonalities between them (i.e. conformationally disordered regions) that may facilitate recognition/degradation by the Bpa-20S system.

Similarly, the authors demonstrate a role for Bpa for bacterial survival during oxidative stress. This phenotype is very strong (even more so than heat shock), but the authors don't follow this up either. I am not sure how these findings with oxidative stress (or even heat shock) fit in the entire story of the manuscript. The authors could have performed proteomic studies also under oxidative stress conditions and try to identify protein(s) whose abundances change in Bpa-deleted bacteria.

Then the story takes a completely different turn; using HspR and two 2 model substrates, the authors try to identify substrate determinants for binding to and degradation by Bpa-20S.

Finally, in Figure 8, the authors proceed once again on a different axis, this time trying to prove that DnaK is a pupylation target and degraded by the Mpa-20S system. Although I understand the authors' intention here (which is to demonstrate a potential interplay between the Bpa-20S system and the Mpa-20S system in regulating the dnaKJgrpE-hspR operon), the experiments in Figure 8 are not sufficient to establish this idea of interplay at a physiologic level. Thus, I feel that the idea in Fig 8 is immature, underdeveloped and probably out of context here.

To sum up, I feel that a clear and focused message is lacking. The authors try to develop this message around the substrate determinants that mediate Bpa binding and 20S-induced degradation, but in this case both Figure 1 and Figure 8 remain as underdeveloped substories that are potentially irrelevant to this central message.

The manuscript primarily investigates the substrate determinants involved in Bpa-dependent degradation, approaching this topic from multiple angles. We have taken into consideration the comment regarding the poor flow of the manuscript text and have made significant revisions to the results section and discussion. These changes aim to enhance readability and emphasize the connections between the different experimental approaches.

Specific comments:

1) In Figure 1, the authors corroborate some of the previous knowledge obtained in *M. tuberculosis* this time in *M. smegmatis*. Specifically, in *bpa* knockout bacteria, they perform proteomic analyses to identify the proteins whose abundances change (when compared with wild type bacteria) both under standard and stress conditions. This experiment shows that HspR accumulates 3-fold in the absence of Bpa even in standard conditions, and 5-fold when cells are exposed to heat shock, suggesting that HspR may indeed be a substrate for the Bpa-20S proteasomal degradation pathway. The authors also report on their discovery of several other proteins whose abundances change upon *bpa* deletion, but they never revisit or follow up any of them.

Because only a few substrates have been identified for ATP-independent proteasomal degradation across species, in my opinion, it would have been interesting to pick another candidate from this list to further study it (i.e. is this protein a novel substrate for Bpa-20S *in vitro*?).

The authors also show that under stress conditions (heat shock: expected; oxidative stress: maybe surprising and novel) the absence of *bpa* is disadvantageous for the bacteria. Because the phenotype with oxidative stress is very strong (total growth impairment in the absence of *bpa*), it would have been interesting to further explore this. For example, would it be possible to perform another proteomic analysis in *bpa* knockout bacteria exposed to oxidative stress to see which proteins accumulate (in comparison with wild type)? This would allow the authors to potentially identify novel Bpa substrates that allow bacteria to survive oxidative stress conditions. They could then check whether this novel substrate also has disordered regions that are essential for Bpa-20S-mediated degradation.

Under heat shock, where the *bpa*-deficient strain shows a phenotype, HspR clearly stands out as the most significant hit, exhibiting the strongest change in abundance and the lowest adjusted p-value. We had, of course, hoped to find other similarly significant hits under this stress condition, but this was not the case. We had, nonetheless, cloned and recombinantly produced some of the less significant candidates. Unfortunately, none of these appeared to be substrates in *in vitro* degradation assays.

We had decided against proteomic comparison of parent and *bpa* knockout strain under oxidative stress conditions, because in our experience, proteome analysis in the context of a severe growth defect usually does not yield useful results. In response to the comment, we have now nevertheless carried out a proteomic comparative analysis of *bpa* knockout versus parent strain as suggested. We observe differences in relative abundance between the wild type and *bpa* deletion strain for a large number of proteins, as is evident from the volcano plot below. This is most likely largely due to their difference in growth phase. We observe significant changes for 845 proteins during oxidative stress (in contrast to the few proteins that change in abundance during heat shock as depicted in Figure 1 B). It would require significant effort and likely a different experimental approach, to identify substrates among the many hits. We therefore consider this to be out of the scope of this manuscript.

[Figure removed by editorial staff per authors' request]

2) In Figure 2, the authors perform in vitro proteasomal degradation assays on HspR, either when it is naked, or after pre-incubation with its target DNA sequence or with its co-repressor DnaK. These assays reveal that HspR, when bound to either of these (likely preventing it from being partially unfolded), is resistant to Bpa-20S-mediated degradation. In addition, here the authors also show that Mpa does not target HspR to 20S, confirming the specificity of Bpa towards this substrate. Although these results are potentially convincing, quantifications of the band intensities (densitometric analyses) from at least 3 such experiments represented in graph would be helpful.

The original experiments were carried out in triplicates. We have carried out the requested densitometric analysis for the replicates and have added the time courses to Figure 2 for easier assessment by the reader.

A

B

3) In Figure 3, the authors aligned the amino acid sequences of 20 different HspR orthologs, and also determined the alpha fold structure of the HspR dimer from *M. tuberculosis*. Eventually, they identified the regions of conservation (in the DNA-binding HTH domain, as well as towards the C-terminus) and predicted that the amino and carboxy termini of the protein contains conformationally disordered regions.

To further study the potential roles of these regions of disorder, they went on to delete them to create the deltaN15 or deltaC9 mutants. The experiments in Figure 4 show that the deletion of the C-terminal disordered region indeed impairs Bpa-20S-mediated degradation. Here, the densitometric analyses are provided and represented on a nice graph.

4) In Figure 5, to investigate how "disordered conformations" actually contribute to degradation, the authors use 2 model substrates either in their native state (RNaseA or FimAa) or locked in their non-native conformations (unfolded alkylated RNaseA or FimAt). As expected, neither the native RNaseA nor the permanently folded FimAa are degraded by the Bpa-20S system, or even by the engineered open gated proteasome. However, surprisingly, unfolded RNaseA or permanently unfolded FimAt are not degraded by Bpa-20S, either. With this observation, the authors suggest that conformational disorder is not sufficient to make a protein a substrate for Bpa-20S. Interestingly, the latter two are actually degraded by the open gated proteasome, suggesting that disorder is in fact required for entry into the 20S chamber. Here my only problem is with Figure 5D. The authors claim that FimAt fused with the C-

terminal stretch of HspR is not degraded efficiently, but this is difficult to see on the gel image. Can they provide quantifications here as well?

We have performed the densitometric analysis and have added the results to Figure 5 D.

5) In Figure 6, the authors perform binding assays using fluorescently labeled Bpa, and aim to test whether the disordered C-terminal part of HspR is involved in Bpa binding. In this assay, they record the binding-initiated changes in the fluorescent intensity, using either WT HspR or the DeltaC9 mutant. Eventually, they conclude that both of these bind with the same affinity to Bpa. Although this is a

beautiful biophysical assay, it would be nice if it was complemented with some robust biochemistry, such as co-immunoprecipitation.

As co-immunoprecipitation is not an equilibrium method, this would not provide us with a way to compare affinities. We therefore believe our biophysical binding assay is better suited to make the statement that both HspR WT and HspR Δ C9 bind Bpa with similar affinities.

6) In Figure 6D, the authors perform rescue experiments to see whether they can restore the degradation of DeltaC9 HspR by fusing it with two different 9 aa long linkers, which do not seemingly help. Looking at their gels, this conclusion is hard to reach. Either better gel images or densitometric quantifications must be presented here, because the bottom two panels actually show that the two linkers indeed promote degradation (maybe not very efficiently, but still to a significant extent). The authors should also clarify what a "GS linker" is.

We agree that in Figure 6D, some level of substrate degradation for the fusion constructs of HspR can be observed. However, the point we make here is that the degradation is considerably slower and the tails fail to restore degradation to that of the wild type HspR. We have performed the densitometric analysis and have added the results to Figure 6 in panel E. The "GS linker" is a nine-residue GS linker with the following amino acid sequence: GGS GSS GSG. We thank the reviewer for noticing our oversight in mentioning the exact sequence of the linker. It is now mentioned in brackets in the text.

7) In Figure 7, by performing a similar binding assay using fluorescently labeled Bpa and 8 short peptides

covering most of the HspR sequence, the authors discover that the region corresponding to amino acids 16-30 mediates HspR binding to Bpa. By performing site directed mutagenesis, they showed that Glu19 and His24 were important for Bpa-20S-mediated degradation.

First off, the authors should clarify why they specifically focused on these two amino acids.

We focused on the charged conserved residues in this stretch, since they pointed outward and did not appear to be involved in stabilization of the helix bundle structure as most of the hydrophobic residues (based on the AlphaFold model). One could argue that arginine 29 should have been included in this analysis. In the revised version of our manuscript, the R29A mutant is now also included.

8) Secondly, they should investigate the binding efficiency (or lack thereof) of H24A, E19K and E19A mutants (either in the context of peptide 1 or in the context of the entire HspR protein) to fluorescently labeled Bpa. This is important and would complement their degradation experiment in Fig 7E. In principle, binding should also be impaired.

We performed the binding experiment with HspR H24A and observed that it binds Bpa less tightly than HspR WT. We have added this analysis to figure S4 as panel F. Unfortunately, variant E19A was not amenable to affinity measurements due to solubility issues.

9) Looking at Fig 7A, degradation of the H24A, E19K and E19A mutants is not fully impaired. How do the authors explain this? Do other amino acids in this stretch compensate for the ablation of H24 or E19? Also, what if the authors combined their deltaC9 mutation with H24A, E19K or E19A? Would that fully abolish degradation?

Mutations in the HspR HTH domain impair but not fully abolish degradation of the substrate by Bpa-CP because it is a cooperative effect of multiple residues interacting with Bpa. Based on the fact that the degradation defect we see for H24A, E19K, E19A, and HspRΔC9 is already significant in the individual variants, we expect that a combination of these mutations would result in full abolishment of degradation. However, due to solubility issues observed already with some of the single residue variants, we have decided not to generate the combined variant.

10) Finally, the authors should test whether the H24A, E19K and E19A mutants are degraded by the

open gated proteasome. In principle, if the point mutations impair binding with Bpa, even the open gated proteasome would fail to degrade the mutants. I think it would be nice to show this.

All variants are degraded by the open-gate proteasome alone, recognition by Bpa is not required. The lack of gating in fact abolished the requirement for the C-terminal tail as threading initiator. Our interpretation is that the overall unfolding/folding dynamics are sufficient. This is further supported by the observation that in presence of the HAIR motif DNA, stabilization against degradation by the open-gate proteasome is achieved. We have added this analysis to figure S4 as panel E (see above) and to figure S5 as panel C.

11) In Figure 8, the authors mention in the text that they have identified (by mass spectrometry) 4 residues on DnaK that are modified by Pup. However, I cannot see any data that support this statement, neither in the main figures nor in supplementary data.

We have added the data as source data.

Minor comments:

Overall, this is an interesting study. As a minor comment, I think that the discussion part is too long and contains a lot of repetitions of the results part. This should be avoided. Also, the authors claim in several places that the C-terminal disordered stretch of HspR is involved in threading towards the 20S core. Although some of the data may suggest this, definitive proof is still lacking, so the authors should try to tone down a little bit.

We have tried to remove repetitions wherever possible and shortened the discussion overall. However, it is difficult to avoid all repetitions, as it is sometimes important to refer again to certain results in the context of other results. We have made sure that our interpretations of the results are careful and warranted and have toned down some of the statements as requested.

Response to Reviewer 3:

Main points:

Deletion of Bpa causes stabilization of the heat shock repressor HspR (Fig. 1). The stress-sensitive phenotype of Δbpa cells can therefore be explained by an insufficient heat shock response, which does not allow synthesis of Hsp70 at sufficient levels (Fig. 9). While this model is appealing, no experimental evidence is provided here. Do the authors observe reduced accumulation of Hsp70 in Δbpa cells after heat shock in comparison to wild type cells?

Under standard conditions we observe a reduced accumulation of Hsp70 in Δbpa cells compared to the wild type of 41% (additional proteomics data uploaded to PRIDE). This is consistent with the finding by Jastrab et al that Hsp70 (DnaK) gene expression is decreased in a *bpa* (*pafE*) deletion strain in *M. tuberculosis* under standard conditions (Jastrab et al., 2015 and 2017).

However, under heat shock, we observe no decrease in Hsp70 accumulation in the Δbpa cells compared to the wild type. The reason likely is that, although Bpa is not present to facilitate proteasomal degradation of HspR, HspR is still not able to re-anneal to the DNA due to it being partially unfolded under the heat stress. Consequently, expression of Hsp70 can still occur. We suggest that under heat stress in a Bpa-deficient strain, Hsp70 is present but tied up in interaction with HspR that accumulates in the *bpa*-deficient strain (model in Figure 8). Less Hsp70 would therefore be available for substrates of the heat shock response (refolding/protecting unstable proteins).

Fig. 2A: The presence of the HAIR DNA operator protects HspR from degradation by Bpa/20S CP. The interaction between HspR and HAIR is believed to be temperature-sensitive, leading to HspR dissociation upon heat shock. Can the authors reconstitute such temperature-dependency *in vitro*? Can HspR be degraded despite HAIR presence at increased temperatures?

We carried out HspR degradations *in vitro* at three different temperatures. At 42°C, HspR is less stabilized against Bpa-mediated proteasomal degradation in the presence of HAIR DNA compared to 30°C and 37°C. However, the observed degradation rate at 42°C is slower than what we observed in the absence of DNA, suggesting that HspR, or at least a fraction of HspR, is still protected by DNA-binding. As the degradation assay is carried out at micromolar concentrations of HspR (4 μ M) and DNA is present at two-fold excess, this is not unexpected. We have added these results to the manuscript as Fig S1B. A quantification by densitometry of the gel time courses that were run in triplicates is provided in Fig S1C.

Fig.3: The sequence determinants of the C-terminal tail, which are crucial for its impact on HspR stability and degradation, remain undefined. According to the alignment hydrophobic and basic residues seem conserved. Can the authors speculate on the relevance of hydrophobicity and positive charge for C-tail function?

Since the C-terminal tail of HspR is also the binding site for Hsp70, the conservation of residues in this region reflects this. The DnaK recognition motif was previously identified as: (basic)-4to5Hb-basic. The sequence ALVVWK in the C-terminal tail of HspR from *M. tuberculosis* fits this recognition motif quite well. To investigate this further, we made point mutations in W121 and K122 (W121A and K122A) as well as the last two residues in the C-terminal tail, R124 and R125 (double mutant R124A R125A). Interestingly, all three mutants exhibited degradation defects compared to wild type HspR. These results are consistent with our hypothesis that the C-terminal region acts as a threading motif with some sequence specificity. We have added these experiments to figure 6 of the manuscript (Fig 6D and E). Fig 6E and F are the densitometric quantifications from gel triplicates performed for Fig 6 D.

Fig.6C: The C-terminal tail reduces the thermal stability of HspR. Can the degradation defects of HspR-ΔC

be compensated by performing the experiments at increased temperatures (e.g. 48 °C) which cause HspR- Δ C destabilization?

We compared degradation speeds of HspR WT and HspR Δ C9 at 30°C, 37°C, 42°C (heat shock). Unfortunately, at higher temperatures (above 45°C) degradation could not be measured reliably due to aggregation. However, we can compare degradation of HspR WT at 37°C with HspR Δ C9 at 42°C, since at these temperatures each protein is 5°C below its melting temperature (T_m HspR WT = 41.6 °C, T_m HspR Δ C9 = 47.8°C). The degradation of HspR Δ C9 at 42 °C is still significantly slower than that of HspR WT at 37 °C. These data suggest that the degradation defect of HspR Δ C9 is not due to changed thermal stability. We have added these results to the manuscript as Fig S3C. Furthermore, we have quantified the degradation speeds densitometrically and added them as Fig S3D.

Fig. 8: The relevance and specificity of Hsp70 (DnaK) pupylation in vitro is questionable. Pupylation of DnaK is extremely slow in comparison to the model substrate PanB and thus not convincing. This result is also not connected to the remaining data focusing on Pup-independent HspR degradation. It is therefore recommended to remove Fig. 8 and the respective section from the manuscript.

The discussion section is very long and sometimes just repeating findings of the results part. A sharpening and shortening of the discussion is therefore recommended.

Since DnaK is part of the HspR operon and is co-regulating the expression of the operon via HspR, we feel that it is an interesting facet that Hsp70 is a pupylation and proteasomal degradation substrate. However, we agree that this should only be taken as a potentially interesting angle to follow up. We also agree that this part is peripheral to the main focus of the manuscript and have moved the figure with these results to the supplement. We have also shortened the section in the manuscript.

We have furthermore made an effort to shorten the discussion and to reduce redundancies, although some referring back to the results section could not be avoided.

July 24, 2023

RE: Life Science Alliance Manuscript #LSA-2023-01923-TR

Prof. Eilika Weber-Ban
ETH Zurich
Institute for Molecular Biology and Biophysics
Otto-Stern-Weg 5
Zurich 8093
Switzerland

Dear Dr. Weber-Ban,

Thank you for submitting your revised manuscript entitled "Pupylation-independent substrate recruitment to proteasomal degradation in mycobacteria". We would be happy to publish your paper in Life Science Alliance pending final revisions necessary to meet our formatting guidelines.

- please address Reviewer 3's remaining comment
- please add ORCID ID for the corresponding author--you should have received instructions on how to do so
- please note that titles in the system and on the manuscript file must match
- please use the [10 author names et al.] format in your references (i.e., limit the author names to the first 10)
- please add a callout for Figure 6F to your main manuscript text;
- you may want to consider uploading Figure 8 as a Graphical Abstract instead of as a figure, but this is up to you
- please make sure the proteomic data is made publicly available at this point

A. FINAL FILES:

B. MANUSCRIPT ORGANIZATION AND FORMATTING:

Sincerely,

Reviewer #2 (Comments to the Authors (Required)):

In their revised manuscript, the authors have now addressed all of my previous comments and concerns and I would strongly recommend publication at Life Science Alliance.

Reviewer #3 (Comments to the Authors (Required)):

In their revised version the authors added new data and clarifications and thus sufficiently addressed my former concerns. There is only one (minor) point remaining: the authors suggest that non-degraded HspR (in Dbpa cells) will titrate Hsp70, preventing the chaperone to bind other cellular substrates and causing a stress-sensitive phenotype. This model will only hold true if HspR accumulates to substantial copy numbers, as Hsp70 (in particular during stress conditions) represents a very abundant protein. It is therefore suggested to provide HspR and Hsp70 copy numbers in stressed Dbpa cells or soften the discussion and respective model (Fig. 9).

Reviewer #3 (Comments to the Authors (Required)):

In their revised version the authors added new data and clarifications and thus sufficiently addressed my former concerns. There is only one (minor) point remaining: the authors suggest that non-degraded HspR (in Dbpa cells) will titrate Hsp70, preventing the chaperone to bind other cellular substrates and causing a stress-sensitive phenotype. This model will only hold true if HspR accumulates to substantial copy numbers, as Hsp70 (in particular during stress conditions) represents a very abundant protein. It is therefore suggested to provide HspR and Hsp70 copy numbers in stressed Dbpa cells or soften the discussion and respective model (Fig. 9).

The reviewer brings up a valid point. Although HspR is encoded in the same operon, it is unlikely to yield similar cellular levels as Hsp70 under stress. We have altered the model (now Fig. 8) and have toned down the respective section in the discussion (lines 598-607).

July 31, 2023

RE: Life Science Alliance Manuscript #LSA-2023-01923-TRR

Prof. Eilika Weber-Ban
ETH Zurich
Institute for Molecular Biology and Biophysics
Otto-Stern-Weg 5
Zurich 8093
Switzerland

Dear Dr. Weber-Ban,

Thank you for submitting your Research Article entitled "ATP-independent substrate recruitment to proteasomal degradation in mycobacteria". It is a pleasure to let you know that your manuscript is now accepted for publication in Life Science Alliance. Congratulations on this interesting work.

DISTRIBUTION OF MATERIALS:

Again, congratulations on a very nice paper. I hope you found the review process to be constructive and are pleased with how the manuscript was handled editorially. We look forward to future exciting submissions from your lab.

Sincerely,
